

# Genome-wide discovery and characterization of long noncoding RNAs in African oil palm (*Elaeis guineensis* Jacq.)

Wei Xia[1,*], Yajing Dou[1,*], Rui Liu[2], Shufang Gong[2], Dongyi Huang[1], Haikuo Fan[2] and Yong Xiao[2]

[1] College of Tropical Crops, Hainan University, Haikou, China
[2] Coconut Research Institute, Chinese Academy of Tropical Agricultural Sciences, Wenchang, China
[*] These authors contributed equally to this work.

Corresponding authors
Haikuo Fan, vanheco@163.com
Yong Xiao, xiaoyong1980@catas.cn

## ABSTRACT

Long noncoding RNAs (lncRNAs) are an important class of genes and play important roles in a range of biological processes. However, few reports have described the identification of lncRNAs in oil palm. In this study, we applied strand specific RNA-seq with rRNA removal to identify 1,363 lncRNAs from the equally mixed tissues of oil palm spear leaf and six different developmental stages of mesocarp (8–24 weeks). Based on strand specific RNA-seq data and 18 released oil palm transcriptomes, we systematically characterized the expression patterns of lncRNA loci and their target genes. A total of 875 uniq target genes for natural antisense lncRNAs (NAT-lncRNA, 712), long intergenic noncoding RNAs (lincRNAs, 92), intronic-lncRNAs (33), and sense-lncRNAs (52) were predicted. A majority of lncRNA loci (77.8%–89.6%) had low expression in 18 transcriptomes, while only 89 lncRNA loci had medium to high expression in at least one transcriptome. Coexpression analysis between lncRNAs and their target genes indicated that 6% of lncRNAs had expression patterns positively correlated with those of target genes. Based on single nucleotide polymorphism (SNP) markers derived from our previous research, 6,882 SNPs were detected for lncRNAs and 28 SNPs belonging to 21 lncRNAs were associated with the variation of fatty acid contents. Moreover, seven lncRNAs showed expression patterns positively correlated expression pattern with those of genes in de novo fatty acid synthesis pathways. Our study identified a collection of lncRNAs for oil palm and provided clues for further research into lncRNAs that may regulate mesocarp development and lipid metabolism.

## INTRODUCTION

Noncoding RNAs (ncRNAs) constitute a critical part of the eukaryotic transcriptome and play a vital role in gene regulation. Long noncoding RNAs (lncRNAs), an important class of noncoding RNAs, are non-protein coding RNAs longer than 200 bp and function as key regulators of diverse mechanisms in a range of biological processes (*Geisler & Coller, 2013*; *Rinn & Chang, 2012*; *Wang et al., 2011*). Researchers have focused on the identification

and characterization of lncRNAs responsible for biological process regulation over the past several decades. LncRNAs were classified into several groups based on their genomic location, including long intergenic noncoding RNAs (lincRNAs), intronic lncRNAs, natural antisense lncRNAs (NAT-lncRNAs) and sense lncRNAs (*Kung, Colognori & Lee, 2013*; *Rinn & Chang, 2012*). Researchers have obtained numerous lncRNAs in plants, including NAT-lncRNAs in *Arabidopsis* (*Zhao et al., 2018*), lincRNAs in black cottonwood (*Shuai et al., 2014*), soybean and wheat (*Golicz, Singh & Bhalla, 2017*; *Zhang et al., 2014*), and lncRNAs associated with stress responses in cotton, grapevine, and Chinese cabbage (*Jain et al., 2017*; *Wang et al., 2019a*; *Wang et al., 2019b*; *Wang et al., 2015*; *Xu et al., 2017*; *Yao et al., 2019*). Oil palm (*Elaeis guineensis*) is an important oil crop in tropical and subtropical areas and its complete genome was released in 2013 (*Singh et al., 2013*). However, no reports on genome-wide lncRNA identification in oil palm are available.

LncRNA loci serve as important regulatory mediators in gene expression and regulate the expression of target genes with *cis*-acting or *trans*-acting mechanisms (*Kung, Colognori & Lee, 2013*; *Liu et al., 2015*). LncRNAs can interact with multiple protein partners, serve as molecular scaffolds that help assemble and target the chromatin-modifying complex, and interact with miRNA as target mimics (*Chu et al., 2011*; *Liu et al., 2012*; *Rinn & Chang, 2012*; *Vance & Ponting, 2014*). For lncRNAs with *cis*-functions, their regulated genes are located close to the lncRNA loci (*Herzog et al., 2014*; *Kim & Shiekhattar, 2016*; *Li, Zhu & Luo, 2016*). NAT-lncRNA loci may regulate their sense genes and influence their expression via diverse transcriptional or post-transcriptional mechanisms (*Zhao et al., 2018*). NAT-lncRNA loci may compete for RNA polymerase II and regulatory transcription factors with their sense genes and cause transcriptional interference (*Faghihi & Wahlestedt, 2009*; *Wight & Werner, 2013*). All these results provided information about screening for lncRNA targets *in silico*. Adjacent genes of lncRNA loci, target lncRNAs of miRNA and complementary sequences between lncRNAs and target genes were used as criteria to select candidate target genes of lncRNAs (*Jiang et al., 2015*; *Li et al., 2015*; *Wu et al., 2013*; *Zhao et al., 2018*).

With the advance of next-generation sequencing (NGS) technology, RNA-seq has become the technical platform of choice to identify lncRNAs. Using bioinformatics tools, lncRNA transcripts can be directly assembled from RNA-seq reads. Recent advances in DNA sequencing and transcriptome analysis have provided gigabases of data and genome-wide analysis of lncRNAs has been conducted in many species (*Jain et al., 2017*; *Ma et al., 2018*; *Wang et al., 2019a*; *Wang et al., 2019b*; *Xu et al., 2017*; *Yao et al., 2019*). Approximately 40,000 putative lncRNAs were identified in *Arabidopsis* by EST, tiling array analyses and RNA-seq data sets (*Jin et al., 2013*; *Liu, Wang & Chua, 2015*; *Wang et al., 2014*). Oil palm is one of the most important oil crops in the world. Many QTL mapping for oil yield, fatty acid composition and vegetative traits in oil palm were reported (*Jeennor & Volkaert, 2014*; *Montoya et al., 2013*; *Singh et al., 2009*), A good reference of oil palm genome, transcriptome profile and lncRNA profile will assist in identifying the causality for phenotypic variations. In this study, we applied strand-specific RNA-seq (ssRNA-seq) technology to identify lncRNA transcripts from an equally pooled RNA sample of oil palm spear leaf and mesocarps of six developmental stages. We also predicted target genes of lncRNAs and analysed the expression patterns of lncRNAs and their target

genes based on 18 oil palm transcriptomes downloaded from the National Center for Biotechnology Information (NCBI) website. LncRNAs expression patterns in different tissues and different developmental stages of mesocarp that stores oil were characterized. We also applied association mapping to identify lncRNA loci related to the variation in fatty acid content. Our study provides a resource for studying lncRNAs in oil palm, including NAT-lncRNAs and their target genes, as well as lncRNAs related to fatty acid content.

## MATERIALS AND METHODS

### Plant materials

One African oil palm plant (hybrid of dura and pisifera) sourced from Malaysia, and grown in the oil palm germplasm resources garden at Wenchang, Hainan, China were used for all experiments. The mesocarp for oil palm fruit at six developmental stages (8-week-old, 12-week-old, 16-week-old, 18-week-old, 20-week-old and 24-week-old), kernel, female flower, and spear leaves were collected from a ten-year-old oil palm individual (accession: CRI-005). The mesocarp at six developing stages and spear leaves were used for the following ssRNA-seq. Roots were collected from one-year-old oil palm seedlings that derived from the ten-year-old oil palm seeds, which were easier to sample and not lignified as the ten-year-old oil palm plant. Each sample collection was done with three biological replicates. All samples collected were immediately frozen in liquid nitrogen and stored −80 °C until needed for RNA extraction. All samples (mesocarp, kernel, female flower, spear leaves, and root) were used for the following quantitative reverse transcription PCR (RT-qPCR) and RT-PCR.

A set of 18 oil palm transcriptomes downloaded from NCBI were also used in this study for gene expression pattern analysis (Table S1). All these transcriptomes were derived from Illumina platform (paired-end, Illumina HiSeq 2000) and included leaf (DRR053156), male flower (DRR053157), leaf from oil palm seedlings (ERR1735779), female inflorescences at 6 leaf stage (SRR5189966), female inflorescences at 15 leaf stage (SRR5189969), root from 3-month-old juvenile tree (SRR7812013, SRR7812014), and mesocarp at different developmental stages, which included ERR1413765 (100 days after pollination (DAP)), ERR1413766 (140 DAP), ERR1413767 (120 DAP), ERR1413768 (160 DAP), ERR3385821 (five months after pollination), ERR3385822 (five months after pollination), ERR3385823 (four months after pollination), ERR3385824 (five months after pollination), ERR3385825 (four months after pollination), ERR3385826 (five months after pollination), ERR3385828 (five months after pollination).

### RNA extraction, RNA-seq libary construction and strand-specific RNA sequencing

Total RNA for oil palm mesocarps (six developing stages described above, with three biological replicates for each stage) and spear leaves used for ssRNA-seq were extracted by a modified CTAB method (Xiao et al., 2012). RNA degradation and contamination was detected by 1.5% agarose gel electrophoresis. Total RNA concentration and purity was determined using a Nanodrop ND-2000 spectrophotometer (Nanodrop Technologies, USA). RNA integrity was verified using RNA Nano 6000 Kit for Agilent Bioanalyzer 2100.

RNA samples without degradation and contamination were used for rRNA removal. The RNA samples were equally pooled and a total amount of 1.5 ug RNA was treated with Ribo-Zero rRNA Removal Kit (epicentre, USA), and then fragmented by fragmentation buffer. First-strand complementary DNA was synthesized using random hexamers and reverse transcriptase. Second-strand cDNA was prepared using RNase H, DNA polymerase I and dNTPs. Remaining overhangs were converted into blunt ends via exonuclease/polymerase activities and 3′ ends of DNA fragments were treated with adenylation. After that, NEBNext adaptors were ligated to the ends of the prepared double-stranded cDNA and the library fragments were purified by AMPure XP beads. The fragment size ranged from 150–200 bp were selected and PCR amplified, and then sequenced with an Illumina Hiseq 2000 platform by Biomarker Technology Co., Ltd (Beijing, China). The clean reads from the above lncRNA-seq data were deposited in the European Bioinformatics Institute (EMBL-EBI) at the European Nucleotide Archive (accession number: ERR3412516).

## RNA transcript assembly and novel transcriptional unit identification

We used the FastQC software (http://www.bioinformatics.babraham.ac.uk/projects/fastqc/) to check the sequence quality of raw reads. Base quality value (Q) was estimated by the following formula: $Q = -10 \times \log_{10} P$ (P represents the error probability during Illumina sequencing). Raw reads were pretreated to remove adaptor sequences and low quality sequences via the Trimmomatic software (*Bolger, Lohse & Usadel, 2014*). The parameters for Trimmomatic were set as: (1) remove adapters (ILLUMINACLIP:TruSeq3-PE.fa:2:30:10); (2) remove leading low quality (LEADING:3), (3) remove trailing low quality (TRAILING:3); (4) scan the read with a 4-base wide sliding window, cutting when the average quality per base drops below 15 (SLIDINGWINDOW:4:15); (5) drop reads below the 36 bases long (MINLEN:36). Clean reads were obtained by removing reads containing adapter, reads containing poly-N and low quality reads from raw data. After raw reads filtering, Q30 value of clean reads was above 95.88% and all reliable reads were mapped to the oil palm reference genome (Version EG5, https://www.ncbi.nlm.nih.gov/genome/2669) using hisat2 (version 2.1.0) with default parameters (*Kim, Langmead & Salzberg, 2015*) and de novo assembled using the StringTie software (version 2.0) with default parameters (*Pertea et al., 2015*). The protein-coding transcripts identified in this study were annotated based on the oil palm gene prediction in NCBI (Gene models based on file downloaded from the NCBI website,https://www.ncbi.nlm.nih.gov/genome/?term=Elaeis+guineensis). After removing all annotated protein-coding genes, transcripts that had more than 200 bp in length, more than one exon and FPKM $\geq$ 0.1 were selected for further identification of lncRNAs.

## lncRNAs identification, classification and characterization

We removed transcripts that were likely to be assembly artifacts according to class code annotated by the gffcompare program and retained transcripts annotated by ''u'', ''i'', ''o'' and ''x'', which represent novel intergenic, intronic, sense-overlapped and cis-antisense transcripts, respectively (*Trapnell et al., 2012*). The transcripts of candidate lncRNAs must not contain open reading frame encoding more than 50 amino acids and must not

encode any transposable elements. Coding Potential Calculator (CPC) (*Kong et al., 2007*), Coding-Non-Coding Index (CNCI) (*Sun et al., 2013*), Coding Potential Assessment Tool (CPAT) (*Wang et al., 2013*) analysis and PfamScan (Pfam 32 database) were applied to analyze transcripts. The assembled transcripts that did not pass the protein-coding-score test (score < 0) of CPC and CNCI analysis were as noncoding sequences. CPAT analyzed the open reading frame length/coverage, Fickett score, and hexamer usage bias of the transcripts and determined the noncoding sequences with default parameter. Based on PfamScan analysis, we eliminated transcript with potential protein-coding ability ($E$-value cutoff $\leq$ 0.001). The predicted long noncoding transcripts shared from the four analyses were considered as candidate oil palm lncRNAs. We filtered out transposable elements for the transcripts through PfamScan analysis and comparing with Dfam database (*Hubley et al., 2015*) through BLAST analysis ($E$-value cutoff <1e−5).

The different types of lncRNAs include lincRNA, intronic lncRNA, anti-sense lncRNA, sense lncRNA were selected using cuffcompare (*Trapnell et al., 2012*). The protein-coding genes were derived from protein-coding transcripts in this study, as well as oil palm predicted gene models downloaded from NCBI (Version: EG5). We used StringTie to calculate Fragments Per Million Fragments (FPKM) values of lncRNAs and other protein-coding genes. For gene with different isoforms, total FPKM values of all isoforms were used to represent the gene's FPKM value.

## Prediction of lncRNA target genes

LncRNAs participated in regulatory pathways through two ways - in *cis* and in *trans*. Target genes for lncRNAs acting in *cis* were predicted by protein-coding genes overlapped within 2 kb flanking sequences of lncRNAs or overlapped with lncRNAs. The genomic positions of lncRNAs and protein-coding genes were compared to identify *cis*-acting target genes for lncRNA. We used the LncTar software to predict target genes in *trans* of lncRNA loci based on mRNA sequence complementary and RNA duplex energy prediction (*Li et al., 2014*).

## LncRNA expression profiles in different oil palm tissues and co-expression analysis

A set of 18 oil palm RNA libraries were downloaded from the NCBI website (https://www.ncbi.nlm.nih.gov/) and used to calculate the fragments per kilobase of transcript per million fragments mapped (FPKM) for lncRNAs and their corresponding target genes (Table S1). Bowtie2 (*Langmead & Salzberg, 2012*) were used to map the reads to oil palm genomes and FPKM values were calculated by cufflinks (*Trapnell et al., 2012*).

Based on the reference of *Zhao et al. (2018)*, Pearson Correlation Coefficients (p.c.c.) were calculated between the expression levels of adjacent protein-coding genes and between the expression levels of lncRNAs and their closest protein-coding genes. LncRNA/protein-coding gene pairs with low abundance ($FPKM_{max}$ <1) were excluded from our analysis. LncRNA/protein-coding gene pairs with Pearson correlation coefficients greater than 0.468 were presented in the heat map ($n = 16$, $P \leq 0.05$). The 18 transcriptome datasets described above were used for analysis of p.c.c. between lncRNAs and genes that belong to the pathways of plastid fatty acid synthesis from pyruvate and triacylglycerol (TAG) synthesis from the reference *Xiao et al. (2019)*.
## Validation of the expression of lncRNAs and their target genes by RT-PCR and RT-qPCR

The total RNAs for oil palm root, stem, spear leaf, female flower, kernel, and mesocarp tissues at six developing stages (8-week-old, 12-week-old, 16-week-old, 18-week-old, 20-week-old and 24-week-old) were extracted by a modified CTAB method as the above. The complementary DNA for each sample were synthesized using All-in-One First-Strand Synthesis MasterMix kit (NOVA, Jiangshu, China) with random hexamers, which was used for RT-PCR and RT-qPCR assays to validate the expression of lncRNAs and their target genes. We used Primer 5.0 to design the primers for these genes and listed the primer information in Table S2.

The RT-qPCR mixture contained 1 μl diluted cDNA, 5 μl of $2\times$ FastStart Universal SYBR Green Master (NOVA, Jiangshu, China), and 0.5 μl of each gene-specific primer (10 μM) in a final volume of 10 μl. All PCR reactions were performed using ABI 7900HT machine under following conditions: 2 min at 95°C, and 40 cycles of 5 s at 95°C and 30 s at 60 °C in 384-well clear optical reaction plates (Applied Biosystems, USA). The procedure ended by a melt-curve ramping from 60 to 95°C for 20 minutes to check the PCR specificity. All RT-qPCR reactions were carried out in biological and technical triplicate. A non-template control was also included in each run for each gene. The final Ct values were the means of nine values (biological triplicate, each in technical triplicate).

## Single nucleotide polymorphisms (SNPs) detection for lncRNAs and identification of lncRNAs related to variation in fatty acid composition by association mapping

Based on 1,261,501 reliable SNPs markers (minor allele frequency >0.05 and more than 80% oil palm individuals had sequences information for each SNP marker) derived from SLAF sequencing in a diversity panel of 200 oil palm individuals in our previous research (*Xia et al., 2019b*), SNPs within lncRNA regions were screened by a Perl script. The contents of lauric acid (12:0), myristic acid (C14:0) , palmitic acid (16:0), tripalmitelaidin acid (16:1), Hexadecadienoic acid (16:2), stearic acid (18:0), oleic acid (18:1), linoleic aicd (18:2), and oil for 160 individual out of 200 oil palm used in this study were the same set of data from our previous research (*Xia et al., 2019a*). Fatty acid composition was examined and measured using gas chromatography (Agilent DB-23, 30 m $\times$250). The nine values (three biological replicates $\times$ three technical replicates) obtained per oil palm individual were averaged for subsequent association mapping. The average values along with standard deviations of different fatty acids and oil contents for the 160 oil palm individuals were listed in Table S3. Since five subgroups for the 200 oil palm individuals were estimated based on cross-validation errors, mixed linear models (MLM) were used. Fixed effects were computed using a Q (population) value matrix, and random effects were computed using a K (kinship) matrix. The Q+K value matrix was added to the MLM model. The Q matrix was obtained using STRUCTURE software (version 2.3.4) (*Pritchard, Stephens & Donnelly, 2000*), and the K matrix (genetic relationships among the 200 oil palm individuals) was obtained using SPAGeDi software (version 1.5) (*Hardy & Vekemans, 2002*). Association

analysis was performed using Tassel 5.0 (*Bradbury et al., 2007*). P-values for associations between SNP markers and fatty acid content were computed according to *Yu et al. (2006)*.

## RESULTS

### Identification and characterization of oil palm lncRNAs

A total of 166,288,038 raw reads were generated from an equally pooled RNA sample of oil palm spear leaf, 8-week-old, 12-week-old, 16-week-old, 18-week-old, 20-week-old and 24-week-old mesocarp tissues, and 165,257,052 clean reads (24.69 Gbp) were obtained after adaptor trimming and sequence quality filtering. Approximately 86.98% clean reads were mapped to the oil palm reference genome, and 96,009 transcripts were assembled. To identify lncRNAs, we filtered out the assembled transcripts shorter than 200 bp and transcripts with protein-coding potential via protein-coding-score test (CPC, CNCI, CPAT) and Pfam protein domain analysis, which identified transcripts with potential protein-coding ability (cutoff $E$-value $\leq$ 0.001). Finally, 1,663 transcripts were tested as having no protein-coding potential and were considered candidate lncRNAs. These transcripts were mapped to 1,363 lncRNA loci and 289 lncRNA loci had more than one transcript (Table S4).

The 1,363 lncRNA loci were distributed across all 16 chromosomes (930) and unlinked scaffolds (433) of the oil palm genome (Table 1). Chr1 (129), Chr2 (96) and Chr3 (95) had the largest number of lncRNA loci, while Chr10 (31), Chr13 (32) and Chr15 (30) had the least number of lncRNA loci. Based on the present annotation version of the oil palm genome, the 1,363 lncRNA loci were classified into four classes. LncRNA loci located in intergenic regions (lincRNAs, 703) are the top number of lncRNA classes, followed by lncRNA loci located on the antisense of protein coding genes (NAT-lncRNAs, 581), lncRNA loci overlapping with pseudogene regions (sense-lncRNAs, 47) and lncRNA loci belonging to genic intronic regions (intronic, 32). Based on the genomic locations for NAT-lncRNA loci and their corresponding sense genes, we further classified them into six types (Fig. 1). NAT-lncRNA loci for type I, II, III, IV overlapped in their genic regions with genes on their opposite strands, while type V and type VI had promoter regions and 3′ UTR regions overlapped with sense strand genes, respectively.

### Expression profiles for identified oil palm lncRNAs

Based on the strand-specific RNA-seq data in this study, the expression levels for lncRNAs and protein-coding genes were compared. A large proportion for lncRNAs (68%) and mRNAs (62%) showed low expression levels (FPKM value <1), while the proportions of lncRNAs (29%) and mRNAs (31%) that had medium expression levels (FPKM: 1∼15) were close (Fig. 2). For genes with high expression levels (FPKM >15), mRNA (7%) had higher gene ratio than lncRNAs (1.6%). Only 23 lncRNA loci had high expression levels (FPKM >15, Table S4).

To investigate the expression pattern of the identified lncRNAs in different tissues, we estimated the expression levels of each lncRNA by calculating FPKM in different tissues using the RNA-seq data from 18 samples for six tissues, including mesocarp (11), female flower (1), female inflorescences (1), male flower (1), leaf (2), and root (2) (Table

**Table 1** Genomic distribution of four lncRNA types across the African oil palm genome.

| Chromosome | LincRNA | NAT-lncRNA | Intronic-lncRNA | Sense-lncRNA | Total |
|---|---|---|---|---|---|
| Chr1 | 61 | 59 | 4 | 5 | 129 |
| Chr2 | 44 | 47 | 2 | 3 | 96 |
| Chr3 | 46 | 46 | 1 | 2 | 95 |
| Chr4 | 38 | 41 | 1 | 4 | 84 |
| Chr5 | 26 | 45 | 1 | 3 | 75 |
| Chr6 | 24 | 31 | 1 | 2 | 58 |
| Chr7 | 21 | 23 | 0 | 4 | 48 |
| Chr8 | 22 | 30 | 1 | 1 | 54 |
| Chr9 | 13 | 33 | 0 | 2 | 48 |
| Chr10 | 11 | 19 | 0 | 1 | 31 |
| Chr11 | 19 | 15 | 0 | 1 | 35 |
| Chr12 | 13 | 28 | 1 | 2 | 44 |
| Chr13 | 19 | 13 | 0 | 0 | 32 |
| Chr14 | 24 | 13 | 0 | 1 | 38 |
| Chr15 | 11 | 16 | 2 | 1 | 30 |
| Chr16 | 12 | 20 | 1 | 0 | 33 |
| others | 299 | 102 | 17 | 15 | 433 |
| Total | 703 | 581 | 32 | 47 | 1363 |

S1). We excluded intronic-lncRNAs and NAT-lncRNAs whose transcripts overlapped with protein-coding genes and used FPKM values for 916 lncRNAs for analysis, which are listed in Table S5. Among these transcriptomes, 77.8% (ERR3385824, mesocarp) - 89.6% (ERR1413765, mesocarp) of lncRNAs had low expression levels (FPKM values 0~1) (Fig. 3), 340 lncRNAs were expressed (FPKM values ≥ 1), and only 89 lncRNA loci had high expression levels (FPKM values ≥ 15) in at least one tissue (Table S5). Among 89 lncRNA loci, 40% (36 loci) had high expression only in one transcriptome followed by 16% (14 loci) in two transcriptomes. There were ten lncRNA loci with high expression levels in more than ten transcriptomes (Table S5 and Fig. 4). The number of lncRNA loci with high expression levels ranged from 12 (ERR3385826, mesocarp) to 36 (SRR5189966, female inflorescences) (Table S5). Comparing 23 loci identified in ssRNA-seq in this study, ten highly expressed lncRNAs (*MSTRG.7397*, *MSTRG.10373*, *MSTRG.26953*, *MSTRG.32440*, *MSTRG.48470*, *MSTRG.51478*, *MSTRG.52125*, *MSTRG.52244*, *MSTRG.53207*, and *MSTRG.53370*) were shared. Seven of the ten lncRNA loci were highly expressed in more than five transcriptomes (Fig. 4).

RT-PCR validation was conducted for 71 lncRNAs among mesocarp, kernel, leaf, female flower and root tissues, and 22 lncRNAs produced positive results (Table S2A, Fig. 5 and Fig. S1). Based on the FPKM values from the strand-specific sequence in this study, 47 out of 71 lncRNAs had relatively low expression (FPKM values ≤ 10), and RT-PCR showed that 9 out of 47 loci produced positive bands in the tested samples (Table S2A, Fig. 5 and Fig. S1). In addition, RT-PCR showed that 10 out of 24 of the tested lncRNAs (FPKM values >10) had positive bands. Among these RT-PCR validated loci, *MSTRG.4577*

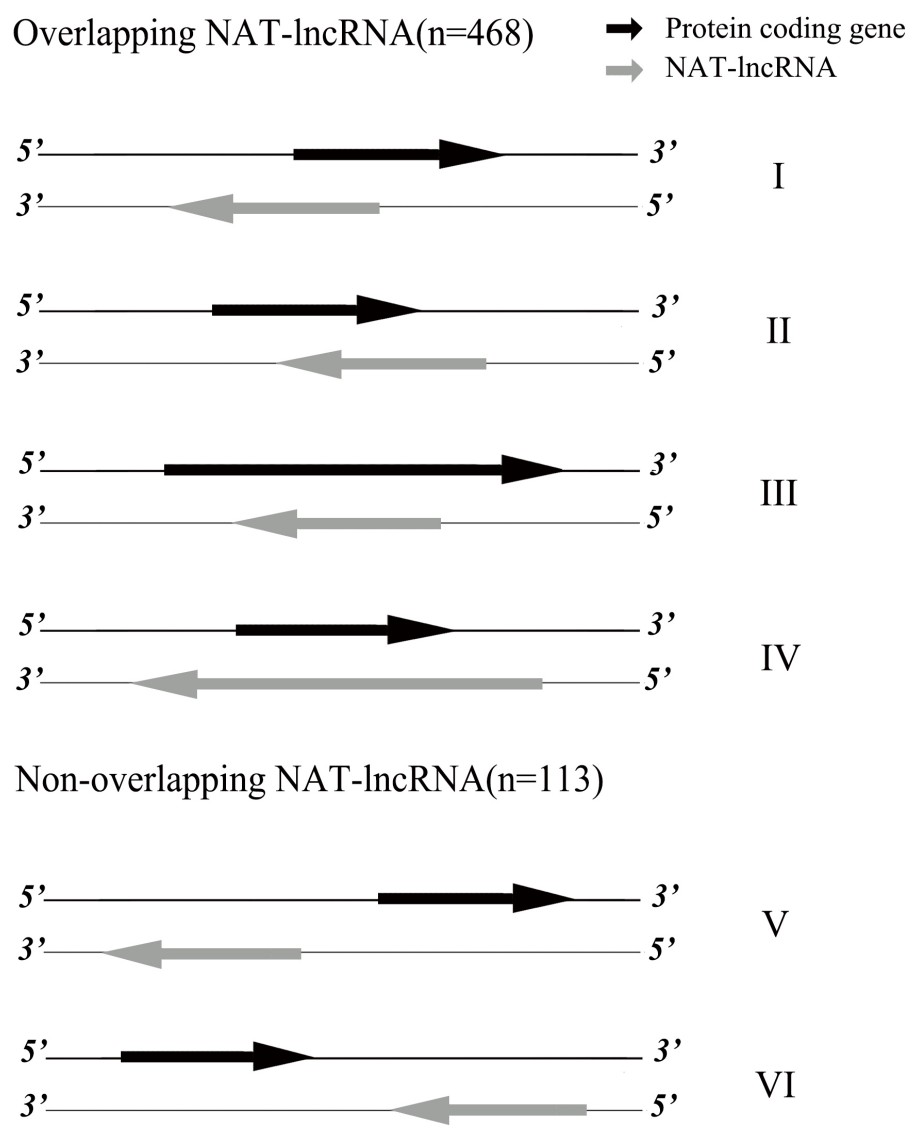

**Figure 1  Classification of NAT-lncRNAs.**

(FPKM = 41.9), *MSTRG.32440* (FPKM = 14737), *MSTRG.48560* (FPKM = 72.8) and *MSTRG.52384* (FPKM = 1.2) were expressed in all five tissues, while *MSTRG.952* (FPKM = 2465.9), *MSTRG.7370* (FPKM = 0.83), *MSTRG.10334* (FPKM = 3.13), *MSTRG.36586* (FPKM = 1.04) and *MSTRG.48470* (FPKM = 49.23) were expressed only in the mesocarp. The remaining 13 lncRNAs were expressed in two to four tissues.

## Target gene identification and coexpression analysis

A total of 875 unique target genes were identified, including natural antisense lncRNAs (NAT-lncRNAs, 712), long intergenic noncoding RNAs (lincRNAs, 92), intronic-lncRNAs (33), and sense-lncRNAs (52) (Table S6). The flanking genes within a 2-kb distance and/or antisense overlapping genes (865) were identified as candidate targets in *cis* for lncRNAs,
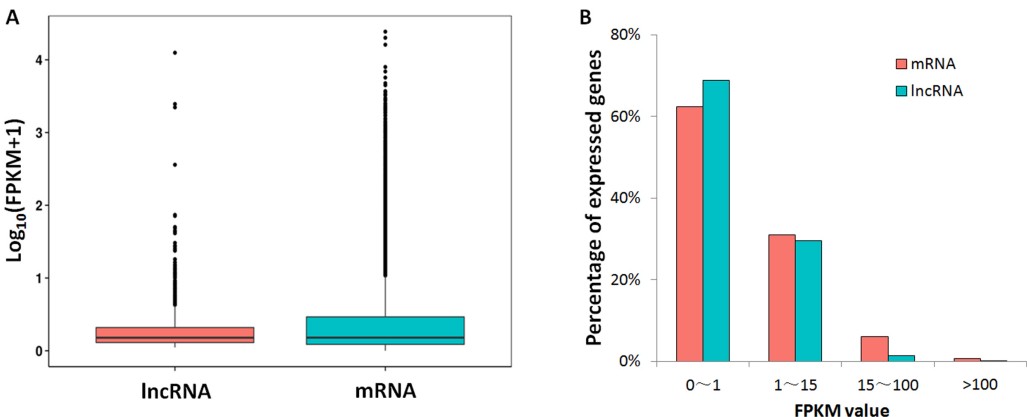

**Figure 2** **FPKM values comparison between lncRNA and mRNA.** (A) Box-plots for $\log_{10}$ transformed FPKM values; (B) lncRNA/mRNA percentages for low expression level (0–1), median expression level (1–15) and high expression levels genes (15–100 and >100).

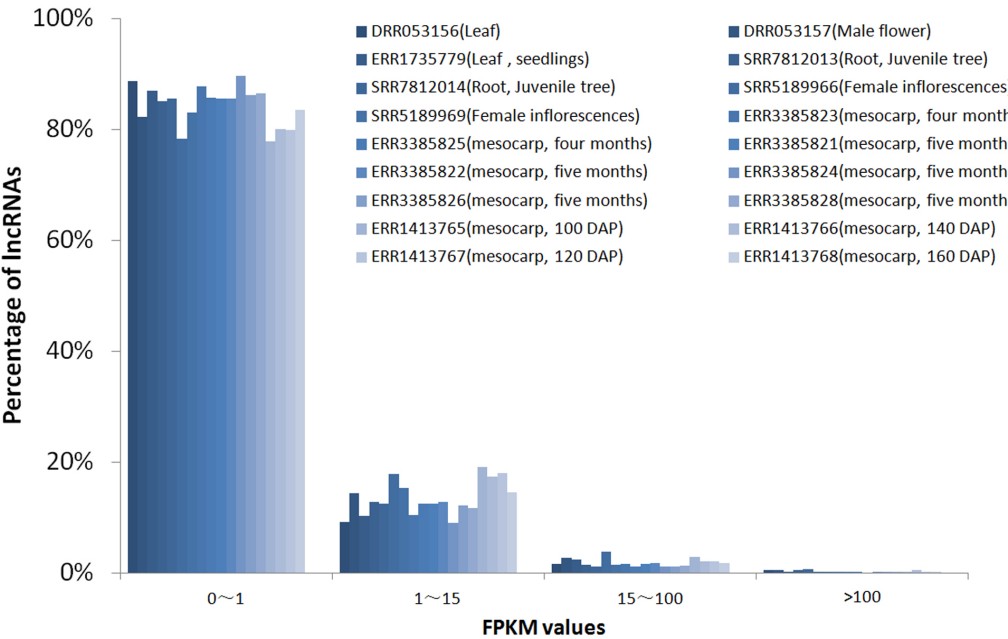

**Figure 3** **FPKM values distribution of lncRNA loci in 18 oil palm transcriptomes.** lncRNA percentages for low expression level (0–1), median expression level (1–15) and high expression levels genes (15–100 and >100).

while 11 predicted target genes were predicted to be in *trans*. One gene LOC105051313, was also identified as a target gene, both in *cis* (*MSTRG.23935*) and in *trans* (*MSTRG.23116*). For NAT-lncRNAs, a total of 712 target genes were found, including target genes overlapping

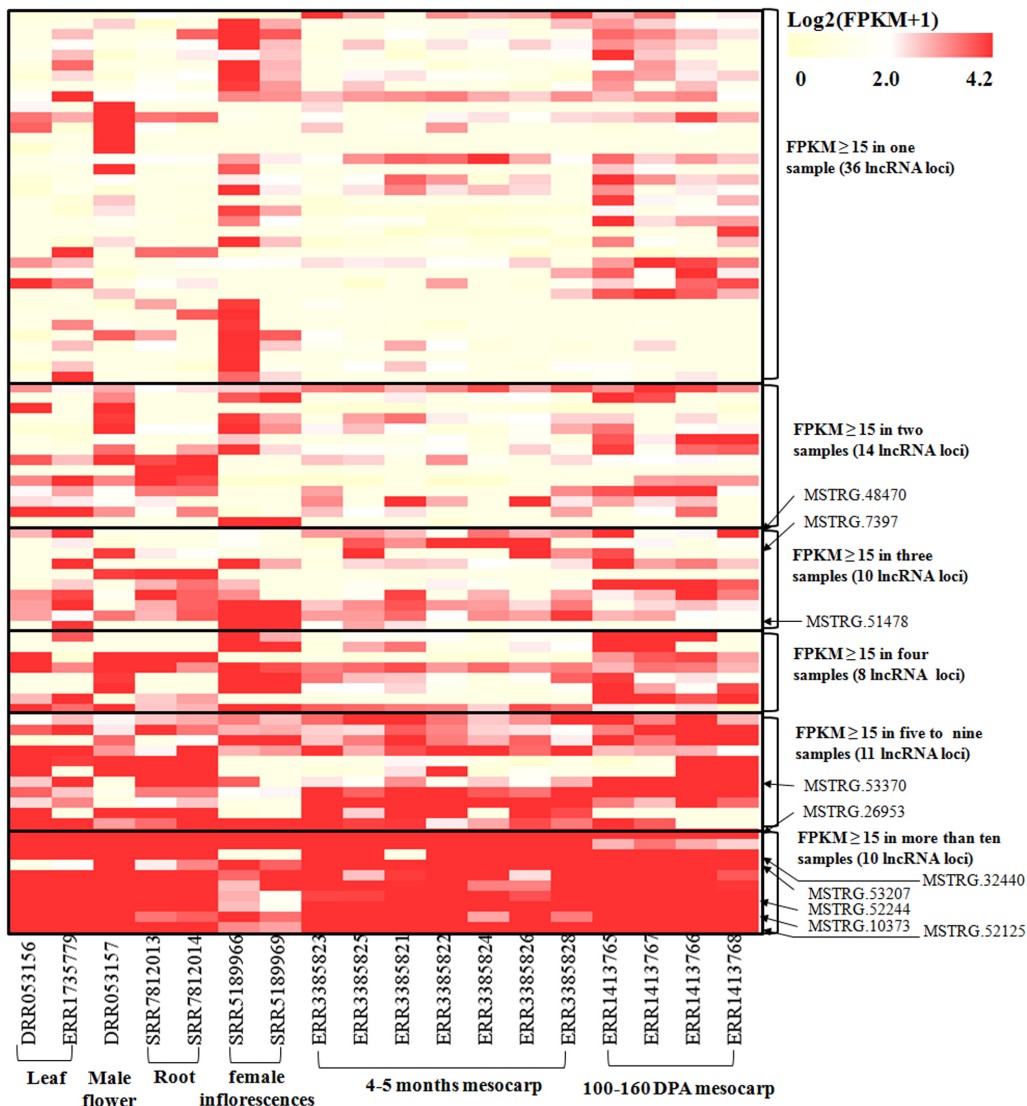

**Figure 4  Heatmap for lncRNAs in 18 oil palm transcriptomes.** FPKM values for each lncRNA was used for K-means clustering and displayed according to the appearance number of high expression level (FPKM ≥15) in 18 transcriptome. LncRNA loci with arrows were highly expressed in both the ssRNA-seq result in this study and in the 18 transcriptomes.

with NAT-lncRNAs on the opposite strand (657), located on the same strand of NAT-lncRNA within the 2-kb region (48), and mRNA sequences complementary to NAT-lncRNA via LncTar analysis (7) (Table S6). Of these target genes, 72% (514) of target genes had genic regions overlapping with NAT-lncRNAs (class I, II, III, IV), and 144 target genes were located within the 2-kb flanking distance of NAT-lncRNAs, including 44 genes overlapping upstream promoter regions with NAT-lncRNAs (class V) and 100 genes overlapping for on the 3′ downstream (class VI). For target genes overlapping with NAT-lncRNA (class I, II, III, IV), comparison of transcripts between NAT-lncRNAs and their target genes indicated that 263 pairs were overlapped and 287 pairs did not overlap

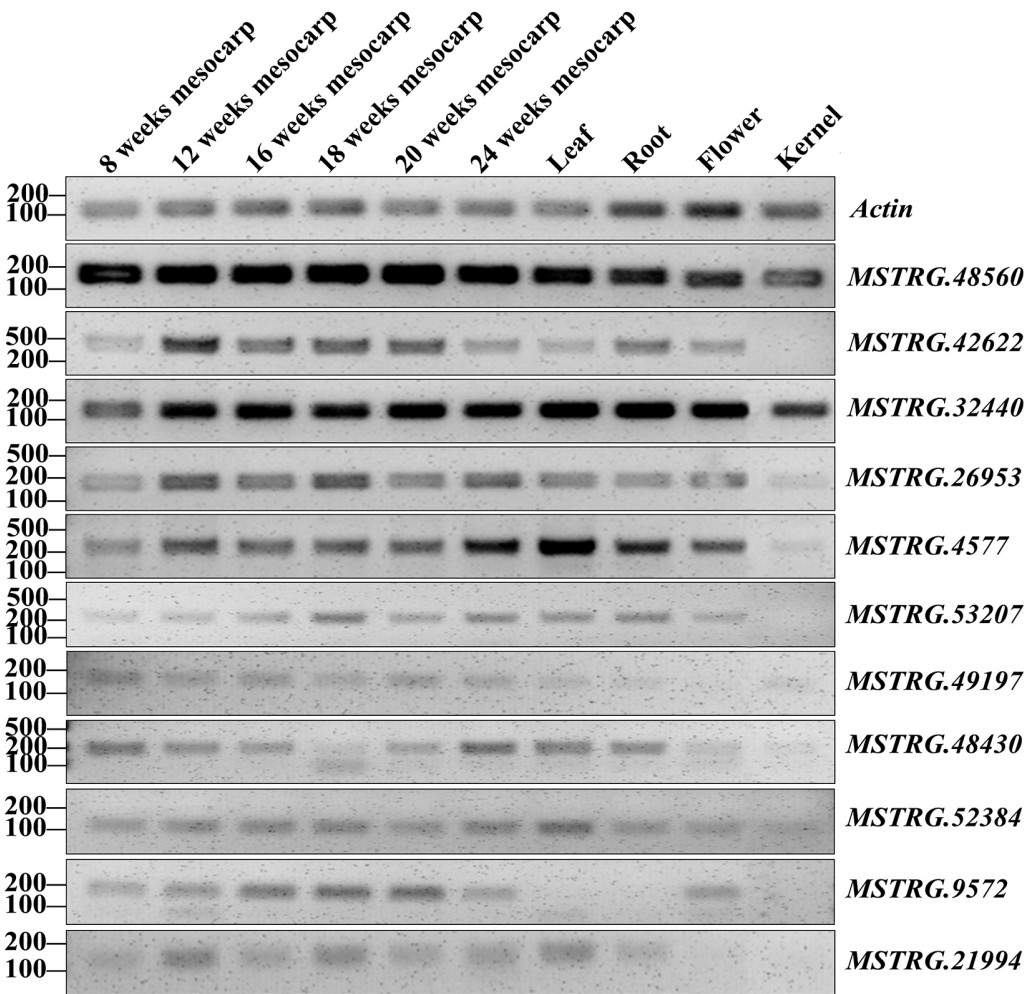

**Figure 5** RT-PCR validation of 11 oil palm lncRNAs in six developing stages of mesocarp, leaf, root, flower and kernel of oil palm.

(Table S5). The lincRNAs (703) were the top number type of identified lncRNAs, and we identified 87 target genes for 82 lincRNAs located within 2-kb regions (Table S6). Among these target genes, eight genes belong to lipid metabolism pathways by comparison with the genes in the pathways identified in our previous research (Table S6) (*Xiao et al., 2019*).

To explore the function of lncRNAs in the regulation of their target genes, we calculated the Pearson correlation coefficients (p.c.c.) between lncRNAs and their target genes. We identified 585 pairs of lncRNA loci and target genes that did not overlap in their transcripts (Table S7). Among these lncRNA/target pairs, 505 lncRNAs had one target gene and 66 lncRNAs had more than one target gene. For these lncRNAs in Table S7, a majority of 312 lncRNAs (250) had low expression ($0 < \text{FPKM}_{max} < 1$) and 84 lncRNAs had no detectable expression ($\text{FPKM}_{max} = 0$) in the 18 transcriptomes, while 47 target genes had low expression and 16 target genes with no detectable expression. Since the existence of low expression levels or no expression among the 585 lncRNAs/target pair, 380 pairs of

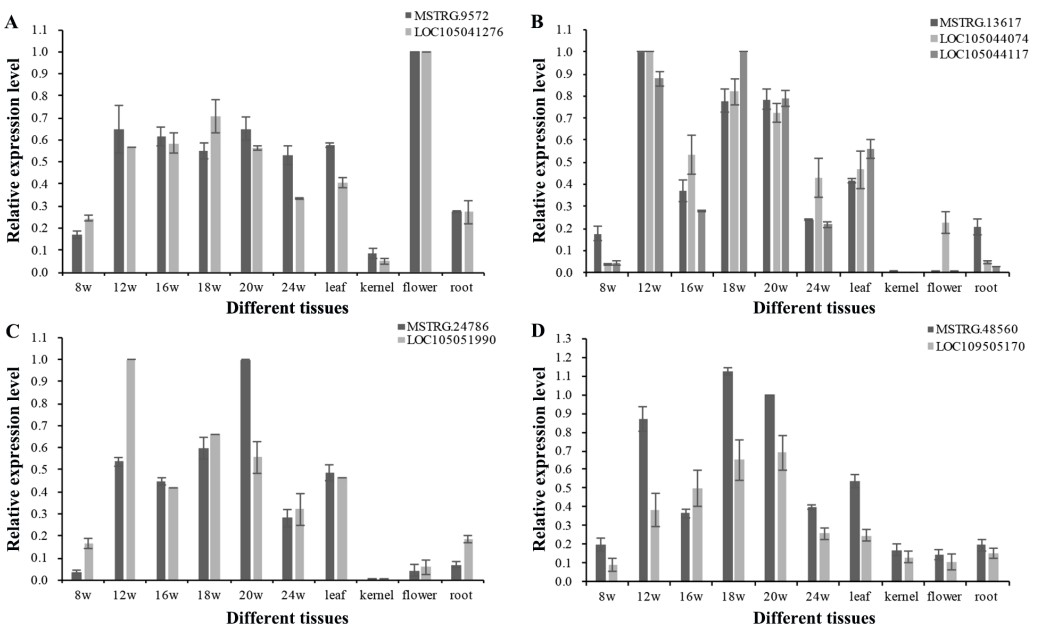

**Figure 6  RT-qPCR validation of four oil palm lncRNAs and their target genes in six developing stages of mesocarp, leaf, root, flower and kernel of oil palm.** Pairs of oil palm lncRNA and their target genes included (A) MSTRG.9572/LOC105041276; (B) MSTRG.13617/LOC105044117 and MSTRG.13617/LOC105044074; (C) MSTRG.48560/LOC105051990; and (D) MSTRG.24786/LOC109505170.

lncRNA loci and target genes were filtered out. The p.c.c. for the remaining 205 pairs of lncRNAs/target genes were examined and 10% (21) of the gene pairs showed positively correlated expression patterns (p.c.c. score ≥ 0.468, *P* value < 0.05) are shown in Table S7. Only one gene pair - *MSTRG.10328/LOC105041936* had negative correlation relationship (p.c.c. score = −0.61, *P* value < 0.05). For the 22 gene pairs, four lncRNAs (*MSTRG.12377*, *MSTRG.18648*, *MSTRG.47162*, and *MSTRG.26385*) and one target gene (LOC105043335) had one transcriptome with $FPKM_{max} \geq 1$, while the other gene in the pairs had $FPKM_{max} \geq 1$ in more than one transcriptomes (Table S7). After RT-PCR analysis for expressed lncRNAs and target genes in the oil palm mesocarp, four NAT-lncRNAs and five target genes proceeded for further RT-qPCR analysis. Positively correlated expression pattern for *MSTRG.9572* / LOC105041276, *MSTRG.13617* / LOC105044074 / LOC105044117, *MSTRG.24786* / LOC105051990, *MSTRG.48560* / LOC109505170 (p.c.c. score >0.9) were detected based on the RT-qPCR results (Fig. 6).

## Genetic variation in lncRNA loci and association with the variation of fatty acid composition

In a previous study, we developed 1,261,501 SNP markers for 200 oil palm individuals (*Xia et al., 2019b*) and determined the contents of lauric acid (12:0), myristic acid (C14:0), palmitic acid (16:0), tripalmitelaidin acid (16:1), Hexadecadienoic acid (16:2), stearic acid (18:0), oleic acid (18:1), linoleic acid (18:2), and oil for 160 out of 200 individuals (*Xia et al., 2019a*). Based on this set of data, 6,882 SNP markers were identified for the lncRNA

**Table 2  SNP markers located within lncRNA regions and significantly associated with the composition of fatty acid in oil palm mesocarp.**

| lncRNA | | SNP position | | Related traits | P-value | Nucleotide types | | | Ave_Het |
|---|---|---|---|---|---|---|---|---|---|
| MSTRG.2123 | Chr1 | 26348240 | Intron | palmic acid content (C16:0) | 0.000794231 | K | G | T | 0.2301 |
| MSTRG.2168 | Chr1 | 27359487 | Intron | Stearic Acid content (C18:0) | 0.000671054 | M | C | A | 0.318 |
| MSTRG.2168 | Chr1 | 27366210 | Exon | Stearic Acid content (C18:0) | 0.000667117 | R | G | A | 0.2188 |
| MSTRG.4816 | Chr2 | 10805479 | Intron | total oil content | 0.000916663 | A | G | R | 0.4067 |
| MSTRG.7396 | Chr2 | 56802941 | Intron | palmic acid content (C16:0) | 0.00038596 | G | R | A | 0.1968 |
| MSTRG.7396 | Chr2 | 56803122 | Intron | palmic acid content (C16:0) | 0.000874185 | T | Y | C | 0.208 |
| MSTRG.7396 | Chr2 | 56767445 | Exon | total oil content | 0.000732102 | G | R | A | 0.1425 |
| MSTRG.7397 | Chr2 | 56767445 | Intron | total oil content | 0.000732102 | A | R | G | 0.2518 |
| MSTRG.10380 | Chr3 | 37185131 | Intron | linoleic acid content (C18:2) | 1.49E−09 | A | G | R | 0.375 |
| MSTRG.11378 | Chr4 | 4103682 | Intron | total oil content | 0.000223078 | T | W | A | 0.1605 |
| MSTRG.11378 | Chr4 | 4103761 | Intron | total oil content | 0.000414367 | T | Y | C | 0.1565 |
| MSTRG.13525 | Chr4 | 50679685 | Intron | total oil content | 0.000982483 | G | A | R | 0.4258 |
| MSTRG.15295 | Chr5 | 25374494 | Exon | linoleic acid content (C18:2) | 0.000862608 | A | R | G | 0.4562 |
| MSTRG.15330 | Chr5 | 27273554 | Intron | total oil content | 0.000838241 | C | T | Y | 0.3712 |
| MSTRG.16341 | Chr5 | 46800615 | Intron | palmic acid content (C16:0) | 0.000916155 | Y | C | T | 0.495 |
| MSTRG.16341 | Chr5 | 46773860 | Intron | linoleic acid content (C18:2) | 0.000914323 | G | A | R | 0.375 |
| MSTRG.16342 | Chr5 | 46773860 | Intron | linoleic acid content (C18:2) | 0.000914323 | G | A | R | 0.3871 |
| MSTRG.16344 | Chr5 | 46800615 | Intron | palmic acid content (C16:0) | 0.000916155 | Y | C | T | 0.495 |
| MSTRG.16345 | Chr5 | 46800615 | Intron | palmic acid content (C16:0) | 0.000916155 | G | R | | 0.3432 |
| MSTRG.17117 | Chr6 | 5087561 | Intron | total oil content | 0.000433608 | T | Y | C | 0.4061 |
| MSTRG.17118 | Chr6 | 5087561 | Intron | total oil content | 0.000433608 | G | R | A | 0.4283 |
| MSTRG.17644 | Chr6 | 21539684 | Intron | total oil content | 0.000343467 | C | T | Y | 0.4793 |
| MSTRG.17644 | Chr6 | 21539697 | Intron | total oil content | 0.000776114 | T | C | Y | 0.4851 |
| MSTRG.28271 | Chr11 | 28838549 | Intron | total oil content | 0.000973214 | A | G | R | 0.4968 |
| MSTRG.28993 | Chr12 | 15680141 | Intron | linoleic acid content (C18:2) | 0.000661354 | Y | T | C | 0.475 |
| MSTRG.28993 | Chr12 | 15680161 | Intron | linoleic acid content (C18:2) | 0.000820421 | R | G | A | 0.4729 |
| MSTRG.29406 | Chr12 | 21510854 | Intron | Stearic Acid content (C18:0) | 0.000778935 | G | R | A | 0.1147 |
| MSTRG.30615 | Chr13 | 13106882 | Intron | total oil content | 0.000815972 | G | R | | 0.1888 |

regions. A mixed linear model was used to analyse the association relationship between these SNP markers and the variation in fatty acid compositions, and 28 SNP markers were significantly associated with the trait variation (cut-off of $1e^{-3}$). One SNP marker (*MSTRG.10380*) had $p$ value lower than $5.01e-08$ ($-\log 10 p = 7.3$) when based on $p$ value threshold in Xia et al. (2019a). These SNPs were located in 21 lncRNAs and associated with oil content (13) and relative contents of palmitic aicd (6), linoliec acid (6), and stearic acid (3). The observed heterozygosity of the 28 SNP markers among 200 oil palm individuals varied from 0.1147 to 0.4968 with an average of 0.3414 (Table 2). A majority of these SNPs (25) occured in the intron regions, while SNPs in *MSTRG.2168*, *MSTRG.7396* and *MSTRG.15295* occured in the exon regions.

To further analyse the correlation relationship between the 21 lncRNA loci and genes belonging to de novo fatty acid synthesis and triacylglycerol (TAG) biosynthesis, 18 transcriptomes from mesocarp (11), female flower (1), female inflorescences (1), male

flower (1), leaf (2), and root (2) were used for calculating p.c.c. between these genes. Seven lncRNAs had positively correlated expression patterns with 35 genes belonging to de novo synthesis of fatty acids (27) and TAG synthesis (8) (Table S8). For the seven lncRNAs, *MSTRG.15295* and *MSTRG.30615* had $FPKM_{max} \geq 1$ in ten and eleven transcriptomes, respectively, while MSTRG.2123 and MSTRG.13525 had $FPKM_{max} \geq 1$ in two transcriptomes and the other three lncRNAs had $FPKM_{max} \geq 1$ in one transcriptome (Table S8). Moreover, a majory of correlated lipid-related genes (29) had $FPKM_{max} \geq 1$ in more than nine transcriptomes. Twenty-seven genes in the de novo fatty acid synthesis pathway, including *EgPDH* (5 genes), *EgACC* (7), *EgACP4* (3), *EgKAR* (1), *EgKASI* (2), *EgKASIII* (1), *EgER* (1), *EgFatA* (1), *EgFatB1* (1), *EgFatB3* (1), *EgSAD-2* (1), *EgHAD* (2), and *EgWRI1-1* (1), were found to be positively correlated with seven lncRNAs. Among these genes, *MSTRG.15295* positively correlated with the top number genes (17) with 15 genes in the de novo synthesis fatty acid pathway followed by *MSTRG.2123* (11), *MSTRG.17117* (9), and *MSTRG.16345* (5). The remaining three lncRNAs were positively correlated with one to three genes positively. *MSTRG.15295* had a positive correlation with stearoyl-ACP desaturase (*EgSAD*) and a transcription factor - *WRINKLED 1* (*WRI 1*).

## DISCUSSION

LncRNAs play important roles in mediating biological process regulation. In this study, we applied ssRNA-seq to identify lncRNAs in oil palm from mixed tissues of leaves and six different developmental stages of mesocarp. Based on ssRNA-seq data in this study and 18 transcriptomes from other studies, we found that the majority lncRNA loci had low expression. Coexpression analysis between lncRNAs and predicted target genes in *cis* indicated that 21 lncRNA loci had positive correlations with target genes, while only one lncRNA locus displayed a negative correlation. Based on the data of SNPs and fatty acid content data from our previous research, 28 SNPs belonging to 21 lncRNAs were associated with fatty acid composition. Our study identified a collection of lncRNAs for oil palm and provided clues for further investigations into lncRNAs that may regulate mesocarp development and lipid metabolism.

Data mining for lncRNAs was feasible since a large amount of transcription data was available. In model plants, such as *Arabidopsis*, maize and rice, a large quantity of lncRNA loci were identified (*Li et al., 2014*; *Zhang et al., 2014*; *Zhao et al., 2018*), some lncRNAs were validated and play critical roles in flowering controls, grain yield, stress response and other biological processes (*Jain et al., 2017*; *Jiang et al., 2019*; *Kindgren et al., 2018*; *Ma et al., 2018*; *Wang et al., 2018*). LincRNAs are the most abundant lncRNA types (Table 1), while the majority of lncRNAs had a relatively lower expression than did protein-coding genes (Figs. 2 and 3). These are common phenomena for lncRNAs in many species (*Wang et al., 2015*; *Yao et al., 2019*). We used mixed samples to conduct ssRNA-seq, which could cover more lncRNA transcripts. The downloaded 18 transcriptomes showed similar low expression levels for most lncRNAs, while lncRNAs highly expressed in more samples tended to be detected in different transcriptomes. The RT-PCR results fit the transcriptome data, 22 out 71 selected lncRNAs were expressed in the tested tissues and four lncRNAs were expressed specifically in the mesocarp.

LncRNAs regulate target genes by serving as target mimics of miRNAs and regulators of transcripion and chromatin modification (*De Lucia & Dean, 2011*; *Jiang et al., 2019*; *Magistri et al., 2012*; *Wang et al., 2018*; *Wu et al., 2013*; *Zhao et al., 2018*). Research in *Arabidopsis* on NAT-lncRNAs demonstrated that NAT-lncRNAs often positively correlate with their cognate sense genes (*Zhao et al., 2018*). Coexpression analysis in this study also suggested that lncRNAs may cotranscribed with adjacent target genes and 6% of lncRNA/target pairs showed a positive correlation in expression (Table S7). The positive correlation results for four lncRNAs and their target genes were also validated by RT-qPCR (Fig. 6). However, since a majority of lncRNA loci had low expression levels, our results suggested that a small proportion of target genes may be cotranscribed with lncRNAs in *cis*.

Oil palm has 90% oil in its mesocarp, the highest level observed in the plant kingdom. *Bourgis et al. (2011)* used RNA-seq to examine transcriptional changes during oil palm mesocarp development, and found that synthesis of fatty acids and supply of pyruvate in the plastid were the major factors controlling oil storage in the oil palm mesocarp (*Bourgis et al., 2011*). Researchers have used QTL mapping and association mapping to identify key loci related to the content or composition of fatty acids in oil palm composition (*Jeennor & Volkaert, 2014*; *Montoya et al., 2013*; *Singh et al., 2009*). In our study, we used association mapping and found 21 lncRNAs related to the variation of fatty acid composition and oil content of oil palm mesocarp (Table 2). The coexpression analysis between the 21 lncRNAs and genes belonging to fatty acid synthesis and TAG synthesis pathways showed that nine lncRNAs had a similar expression patterns as genes in the two lipid metabolism pathways. *MSTRG.15295* had similar expression pattern with the majority of correlated genes, including a key transcription factor - *WRI1*. The *WRI1* gene is considered to play an important role in oil accumulation (*Bourgis et al., 2011*; *Cernac & Benning, 2004*; *Dussert et al., 2013*; *Focks & Benning, 1998*; *Troncoso-Ponce et al., 2011*). Our results provide candidate lncRNA loci related to the development and oil storage of oil palm mesocarp.

## CONCLUSIONS

This study provided an important collection of lncRNAs in oil palm and a set of 581 NAT-lncRNAs and 712 targets were predicted. Based on 18 oil palm transciptomes, we found that A 77.8%–89.6% lncRNA loci had low expression, while only 89 lncRNA loci had medium to high expression in at least one transcriptome. Coexpression analysis between lncRNAs and their target genes indicated that 10% of lncRNAs had positively correlated expression patterns with target genes. Based on SNP markers derived from our previous research, 28 SNPs belonging to 21 lncRNAs were associated with the variation of fatty acid contents. Moreover, twelve lncRNAs showed positively correlated expression patterns with genes in de novo fatty acid synthesis pathways. Our study identified a collection of lncRNAs for oil palm and provided clues for further research investigating lncRNAs that may regulate mesocarp development and lipid metabolism.

### Funding

This work was supported by grants from the National Natural Science Foundation of China (No. 31301358), the Scientific and Technological Cooperation Projects of Hainan province (No. KJHZ2015-06) and the Fundamental Scientific Research Funds for Chinese Academy of Tropical Agricultural Sciences (No. 1630152019006). The funders had no role in study design, data collection and analysis, decision to publish, or preparation of the manuscript.

### Grant Disclosures

The following grant information was disclosed by the authors:
National Natural Science Foundation of China: 31301358.
Scientific and Technological Cooperation Projects of Hainan province: KJHZ2015-06.
Fundamental Scientific Research Funds for Chinese Academy of Tropical Agricultural Sciences: 1630152019006.

### Competing Interests

The authors declare there are no competing interests.

### Author Contributions

- Wei Xia conceived and designed the experiments, analyzed the data, prepared figures and/or tables, authored or reviewed drafts of the paper, and approved the final draft.
- Yajing Dou performed the experiments, analyzed the data, prepared figures and/or tables, and approved the final draft.
- Rui Liu and Shufang Gong analyzed the data, prepared figures and/or tables, and approved the final draft.
- Dongyi Huang conceived and designed the experiments, authored or reviewed drafts of the paper, and approved the final draft.
- Haikuo Fan and Yong Xiao conceived and designed the experiments, analyzed the data, authored or reviewed drafts of the paper, and approved the final draft.

### DNA Deposition

The following information was supplied regarding the deposition of DNA sequences:
The clean reads from lncRNA-seq in this study are available in the European Bioinformatics Institute (EMBL-EBI) (https://www.ebi.ac.uk/ena/submit/sra/#home) at the European Nucleotide Archive: ERR3412516.
https://www.ebi.ac.uk/ena/data/view/ERR3412516.

### Data Availability

Data is available at EMBL-EBI ENA: ERR3412516.

### Supplemental Information

Supplemental information for this article can be found online at http://dx.doi.org/10.7717/peerj.9585#supplemental-information.

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
