# Peer review of "Genome-wide discovery and characterization of long noncoding RNAs in African oil palm (Elaeis guineensis Jacq.)"

_PeerJ, doi:10.7717/peerj.9585_

## Round 0.1 · original submission · Major Revisions

Three reviewers have identified multiple issues with both grammar and scientific content of your manuscript. There is a need for the manuscript to be carefully proof-read, by a native English speaker or a professional editor. In addition, the bioinformatics methods are not well described and are not commonly accepted, standard methods. If you are proposing a novel bioinformatics pipeline, it should be carefully described and justified. I suggest that you address all concerns by the reviewers and resubmit the improved manuscript.

Reviewer 1 ·

Basic reporting

This paper provides a characterisation of lncRNAs in oil palm, using both novel and public RNA-seq libraries (some produced by the same group for an earlier publication, but not necessarily all – this is not, at present, clear). The combination of multiple RNA-seq datasets and the conservative approach used to identify candidate lncRNAs (being the intersection of 4 different coding potential predictors) ensures that as a catalogue of novel transcripts, the paper contributes a tranche of useable data. However, I have several concerns related to presentation and methodology.
Most notably, the distinction between the datasets generated solely for this manuscript and those generated elsewhere should be made clearer as the ‘plant materials’ section of the methods (lines 100-101) combines the two – but not all of the samples stated there are sequenced using methods described in the present manuscript. Most obviously, the mesocarp samples detailed in Table S1 (sourced from BioProject PRJNA66349) are sequenced using a 454 machine, not Illumina. Accordingly, it becomes confusing as to what data is being used for what purpose, a problem aggravated by what are presumably typos in the number of transcriptomic datasets used: I believe it should be 22 (line 33), but at other points in the paper it’s 11 (line 279), 23 (legend of figure 3), 26 (line 223), and 32 (line 149).

There are also numerous, albeit minor, errors in spelling and grammar throughout. I’ve identified some of the more obvious, but would recommend a more thorough proofreading.

Line 34. As a matter of style here and throughout the rest of the manuscript, the thousands separator should be a comma, not a period (as used on line 185).

Line 35. ‘targets gene’ should be ‘target genes’

Line 48. ‘contain’ should be ‘are’

Line 61. ‘identification’ should not be capitalised.

Line 65. ‘populus’ should either be the full species name (Populus trichocarpa) or the common name (black cottonwood).

Line 66. The phrase ‘and so on’ should be avoided, as it implies but does not provide additional citations to support the point. See also line 74.

Line 68. ‘reseased’ should be ‘released’. ‘area’ should be pluralised. At the start of the sentence, there appears to be a missing citation between a pair of empty parentheses.

Line 72. ‘transacting’ should be ‘trans-acting’

Line 75. ‘regulated genes locate close to’ should be ‘regulated genes are located close to’

Line 80. ‘in silico’ should be italicised.

Line 85. ‘gigabase data’ should be ‘gigabases of data’ (although this vastly understates the amount of publicly available RNA-seq data: the SRA is many petabytes in size).

Line 88. ‘Arabidopsis’ should be italicised.

Line 90. ‘mixture tissues of oil palm leaf and mesocarps of six developing stages’ is unclear: if tissue samples are not explicitly being pooled, ‘mixture’ would be better phrased as ‘range of’. It would also be clearer to state which tissues were actually sourced for lncRNA assembly (is the word ‘leaf’ misplaced here – line 101 states that root and stem tissues were also collected, but were these sequenced for this purpose? My interpretation of line 197, which again refers only to leaf, was that they were not. So why are they mentioned in the methods alongside those tissues that were?) ‘Developing stages’ should be ‘developmental stages’; it would also be clearer to give the range (8-24 weeks).

Line 91. ‘deep sequencing of 16Gb data’ is not a clear phrase. I read this as ‘deep sequencing producing 16Gb of data’, but would note it’s uncommon to refer to file size in this context. This also contradicts line 198, which gives a file size of 26.49Gb (consistent with that uploaded to the ENA).
Given the word ‘deep’, it would be far better to omit file size and provide an approx. read number (line 197 suggests it’s c. 20m per sample for 8 samples) or estimate of coverage (i.e. how deep is deep? 5-fold? 10-fold?).

Line 91. ‘together with 22 oil palm transcriptomes’. As this data is being sourced from the public domain, the associated publications should be cited at this point. It isn’t immediately obvious how many other studies have been used, and whether they are all from the same research group (albeit used for a previous manuscript). This is because while Table S1 provides the SRA run accession, it does not provide the corresponding (group-level) BioProject accession, nor citation.

Line 94. ‘colleration’ should be ‘correlation’.

Line 117. ‘illumina’ should be capitalised.

Line 126. ‘de novo’ should be italicised.

Line 132. ‘fram’ should be ‘frame’.

Line 135. ‘analysis transcripts’ should be ‘analyse transcripts’.

Lines 138 and 187. ‘perl’ should be ‘Perl’.

Line 139. ‘calcalated’ should be ‘calculated’.

Line 144. ‘cis’ and ‘trans’ should be italicised, here and throughout the rest of the manuscript. However, it should be clarified that this manuscript only considers candidate cis-acting target genes.

Line 149. By reference to line 91 and table S1, this figure of 32 should be 22. On line 223, the figure of 26 should also be 22. In the legend of Figure 3, the figure of 23 should also be 22.

Line 152. ‘used to mapped’ should be ‘used to map’.

Line 210. ‘pesodogene’ should be ‘pseudogene’.

Line 270. ‘variaton’ should be ‘variation’.

Line 281. ‘expreesion’ should be ‘expression’.

Line 289. 'potitively correrated' should be 'positively correlated'.

Experimental design

One of my principal concerns is that the informatics methodology is not yet completely described, with some analyses given only cursory detail.

For example:

Homology searching was (presumably) performed to ensure candidate lncRNAs do not encode transposable elements (line 133), but there is no more detail than this.

The rightmost column of Table S2 states, for some genes, ‘enhancer predicted.’ By what method was this prediction made? The word ‘enhancer’ does not appear elsewhere in the manuscript.

Line 127 specifies that ‘all annotated and pathway identified gene were removed’ but does not state how genes were assigned to pathways (although presumably the annotation was sourced from NCBI: ftp://ftp.ncbi.nlm.nih.gov/genomes/all/GCF/000/442/705/GCF_000442705.1_EG5/GCF_000442705.1_EG5_genomic.gff.gz? Clarification would be helpful). There is, I believe, methodological overlap with the previous Xiao et al paper – if the pathways were identified using methods detailed there, it would be helpful to provide a brief summary.

At other points the methods are somewhat atypical:

On line 123, raw reads were filtered using (presumably) in-house scripts, an uncommon choice given the number of tools that do this and the conventions they follow. As read quality degrades towards the 3’ end, it is common practice to trim bases below a minimum Phred score, but this was not attempted – rather, reads were dropped if > 50% of their bases had Phred < 10. Given most of the low-quality bases will be towards one end (and won’t, in a correctly prepared library, constitute half the read length anyway), is it likely this filter is doing anything substantial? What % of reads were filtered out on this basis? Were the adapters (which ones?) identified by exact matching and how were adapter-containing reads treated: were they discarded if they contained a match or was the adapter sequence excised?

In general, it isn’t particularly clear what sequencing data is being used for what purpose, with some of the analyses in this study using data from a related publication (Xiao, et al.). For the lncRNA assembly described in this paper, line 197 states that 8 samples were sequenced: mesocarp at 6 time points, and ‘leaf’ (presumably, 2 replicates). This data is deposited in the ENA, as stated on line 120. However, the corresponding BioProject for this paper (https://www.ebi.ac.uk/ena/data/view/PRJEB33304) contains only one run accession, so were all 8 samples pooled together or is the publicly available data only a fraction of the whole? The metadata for this run (SRA RunInfo table) does not specify the tissue of origin, nor for that matter, whether it is a pool of tissues: the "Sample_Name" field is just "lncNRA".
Furthermore, the ‘plant materials’ section (lines 100-101) refers to kernel tissues, root, stem and pollen. If these samples were originally collected for a previous publication, this should be clearly identified here.

For reproducibility, version numbers and parameters should also be provided for all tools (if the default parameters were used, this should also be stated) as in places the methodology is otherwise the simplistic, e.g., ‘using Hisat2’.

I have reservations about the co-expression analysis and would not be able to resolve them without a table of expression level data.
Meaningful correlations between expression profiles can only be made if, to begin with, each of the 22 RNA-seq libraries (detailed in Table S1) is generated using a common method. However, as this data was not generated specifically for this study, the methods are not detailed in the paper, so it is not immediately clear that this is the case. Line 224, and Figure 3, suggests that expression levels across these 22 libraries (which represent 6 tissues) are averaged to give one FPKM estimate per gene per tissue. (Which would collapse the mesocarp developmental stages together, although I note this was not the case for Figure 6).
If so, my concern is that Pearson’s correlation coefficients can be highly skewed by outliers, which is very likely given the low number of datapoints (n=6) and the sparsity of many lncRNA expression profiles (it is expected that of the 6 datapoints per gene, many will be 0; the characteristically low expression of many lncRNAs is already evident from Figure 1B). Notably, the filter criteria described on line 156 (excluding lncRNAs where the maximum FPKM, across all 6 datapoints, is < 1) would not resolve this problem.
Spurious correlations would affect some of the target gene relationships drawn by this paper (e.g. that of MSTRG.17644 and WRINKLED1, on line 287) but in the absence of expression level data, these cannot be scrutinised.

Differential expression analysis is also cursorily described, on line 228, as being a fold change > 2 between any two tissues. Again, in the absence of expression level data (and a minimum expression threshold), these cannot be meaningfully interpreted. At very low FPKMs, transcription is essentially indistinguishable from noise, artificially exaggerating the effect of fold change, e.g. there’s a 10-fold difference between 0.1 FPKM and 0.01 FPKM, but it is not reasonable to believe this is all that biologically relevant nor constituting ‘differential expression’ in any substantive sense.

Validity of the findings

See above comments on experimental design, particularly the provision of expression level data. It is not possible to adequately assess the validity of findings in its absence.

·

Basic reporting

The manuscript needs careful editing for English spelling and grammar. The title and abstract have misleading content that are not supported by the data included in the manuscript.

Experimental design

There are weaknesses in the data for the association part of the study (see detailed comments below).
Method details are missing for which samples were used in which analysis and for the criteria for qualifying lncRNA and targets (see detailed comments below).

Validity of the findings

1. Title needs revision as “Fatty acid content” has not been measured – only gene expression of FA biosynthetic pathway transcripts, the title is very misleading
2. Line 31 “In this study, we applied strand-specific RNA-seq with rRNA removal to identify lncRNAs from two oil palm tissue types - leaf and six different developing stages of mesocarp” but in introduction Line 91 says “Together with 22 oil palm transcriptomes for leaf, root, pollen, flower, mesocarp and kernel tissues, we systematically identified and characterized oil palm lncRNAs…”
3. It is also unclear in methods which tissues were used in which parts of the study (all mentioned in plant materials (line 98) but only leaf and fruit mentioned in RNA extraction (line 106) some were only used for the RT-PCR, but its not clearly presented as there are two methods for “RNA extraction” here (diff methods for PCR work Line 167)
4. Line 68 ( ) (did you mean to include the scientific name here?)
5. Line 79 “all these results will provide” ? this is lit review should use past tense (grammar error)
6. Line 98 “dura” is not the name of a single cultivar, but is a general term used for palms with a particular fruit phenotype (thick shell) and also wild type palms from Africa. Palms with “dura” phenotype fruit are used at mother palms for breeding and are not used in commercial oil palm fruit production.
7. For all methods, please use past tense (not present tense)
8. Line 125, please provide URL for the oil palm reference genome
9. Line 132 typo “open reading fram”
10. Line 133 “Coding Potential Calculator (CPC) (Kong et al. 2007), Coding-Non-Coding Index (CNCI) (Sun et al. 2013), Coding Potential Assessment Tool (CPAT) (Wang et al. 2013) analysis and PfamScan (Pfam 32 database) were applied to analysis transcripts, and predicted long non coding transcripts shared from the four analysis were considered as candidate oil palm lncRNAs.” What were the criteria / score cutoffs used to determine qualifying lncRNA sequences? (similarly, for line 202, explain what is required for “transcripts (that) passed all four analyses”)
11. Line 144 “LncRNA participated in regulatory pathways through two ways - in cis and in trans. Target genes for lncRNAs acting in cis were predicted by protein-coding genes overlapped within 2kb flanking sequences of lncRNAs or overlapped with lncRNAs. Perl script was programmed to identify cis-acting target genes for lncRNA.” So were only cis-targets identified in this study? Has the Perl script (here and also Line 187 for SNP method) been provided with the submission?
12. Line 149 define “SRA”
13. Line 159 “The transcriptomes used for analysis p.c.c. between lncRNA and genes belongs to the pathways of plastid fatty acid synthesis from pyruvate and TAG synthesis the reference” remove "the reference"
14. Line 192 “P- values for associations between SNP markers and fatty acid content were computed according to Yu et al (2006).” Where is the FA content data? (no method to get that data here and no ref to where the data is)
Results
15. Line 255 typo “Among these targets genes, 21 genes belong to with lipid (8) and carbohydrate metabolism pathways (13) by comparison with the genes in the pathways identified in our previous research”. Also please clarify what “carbohydrate metabolism” pathways mean there as the 21 genes (including the 13 “carbohydrate metabolism” genes) are later all refereed to as associated with lipid synthesis later in the manuscript
16. Line 256 “Sixteen genes with high expression in mesocarp were listed in Table 2 and all these genes were targeted by NAT-lncRNAs.” Clarify that there is no functional evidence – these are only predicted targets
17. Line 259 -268 As different types of lncRNA have different mechanisms of action, and again, as these predictions can only be a probably biased subset of the real lncRNAs and targets, I don't see the usefulness of the correlation analysis
18. Figure 5 is not at all informative: It is not possible to view any association with direction of expression between any lcnRNA and its predicted target from this figure.
19. Line 270 typos “Genetic variaton in lncRNA loci and association with the variation of fatty acid compostion”
20. Line 271 typo "previoius"
21. Line 271 – 277. No fatty acid content data is provided. No data for analysis of SNP effect on lncRNA structure or lncRNA binding efficiency is provided. Any association without this supporting data is not robust.
Discussion
22. Line 300 “28 SNPs belong to 21 lncRNAs were associated with fatty acid composition” there is insufficient data to validate this claim
23. Line 311 typo “Clustering analysis of lncRNAs expression also suggested that lncRNAs were develomentally regulated (Figure 3)”. Since the data sets for different tissue were from very different palm materials (some dura, some pisifera, some tenera) and different genetic backgrounds, this is again, very tenuous, but I would agree it can be said it is something worthy of further (and more robust) study.
24. Line 316 “Coexpression analysis in this study also suggested that lncRNA may co-transcribe with adjacent target genes and 25% lncRNAs-targets showed positive correlation in expression (Figure 5).” See comments on results for this. Also, there is no discussion of what this 25% means functionally or why the authors think this is useful information.
25. Line 325 “In our study, we used association mapping and found 21 lncRNAs related to the variation of fatty acid composition and oil content of oil palm mesocarp (Table 3).” See comments on results – the data are insufficient to make this statement valid.
26. In general, the data and results need to be revised then the discussion and conclusions revised accordingly.

Additional comments

The manuscript provides a useful preliminary and predictive analysis of lncRNA in oil palm, however, the title and abstract are misleading and are not justified based on the data provided in the manuscript: As the “fatty acid content” of these plant materials has not been determined and no data for fatty acid levels is included in the manuscript, it is incorrect to conclude any association with fatty acid content. The “variations” here can only refer to variation of transcript abundance of fatty acid biosynthetic genes that are predicted to be targets of some of the lcnRNA. While the first part of the study (analysis and prediction of lncRNA and some of their targets) is acceptable as a preliminary and predictive report, there is no functional data to support the claims made for the remainder of the manuscript. A major weakness is that the prediction of lncRNA targets is limited to those that are in cis to their lncRNA (as there are no reliable predictive methods for those functioning in trans, other that miRNA mimics), so the data is bound to be highly noisy as many of the relevant gene transcripts (for FA synthesis) have to be excluded and cannot be used as comparators, as it is unknown whether or not they are targets of lncRNA. For the same reason, any apparent association of predicted lncRNA target transcripts with SNPs in any of these predicted lncRNA is preliminary because, while the RT-PCR data can be used to validate the transcript level, we still don't know if this is the case for all lncRNA targets, as these are only the targets that can be predicted (and we can’t know if there are others) and there is a non-validated assumption that these genes qualify to represent lncRNA regulation. There is also no attempt to explain if or how the SNPs affect lncRNA function: Are they in positions that change the folding? or do they affect the binding energy with complementary regions (for those that function by complementarity)? The presence of SNPs within lncRNA has already been published by this group in the paper that they cite, but these SNPs cannot be extrapolated to any functional association with fatty acid content because there is insufficient supporting data: the lncRNA are only predictions, the FA biosynthesis genes mentioned are only the predicted targets of predicted lmcRNA, and the fatty acid content variation is also only a prediction based on PCR data for a some of these predicted targets (i.e. a prediction based on a prediction that was based on a prediction with no functional validation).

Reviewer 3 ·

Basic reporting

In general, the authors need to proofread the manuscript again. Many grammatical errors, incorrect spellings and inconsistent terminologies are spotted. Some sentences are really confusing. Perhaps, the English language needs to be further improved to ensure good understanding of the international readers. Some examples are given below. There are more and I cannot mention all of them here.

1. Line 30: Please change to ‘noncoding’, instead of ‘non coding’.

2. Line 34-36: ‘A total of 865 target genes for 35 NAT-lncRNA, lincRNAs, intronic-lncRNAs were predicted, including 581 NAT-lncRNAs and their targets gene with high confidence.’ Suggestion: this sentence should be more clear. Without reading the results, the ‘high confidence’ here is clueless. The authors may consider changing to ‘the coexpression between 581 NAT-lncRNAs and their target genes were found to be significant’. Besides, lncRNAs with plural ‘s’ is also inconsistent. Small mistakes like ‘targets gene’ should be avoided.

3. Line 72: It should be ‘trans-acting’, instead of ‘transacting’.

4. Line 75: The authors should differentiate the lncRNAs from the loci producing the transcripts. To avoid from confusion, lncRNA locus is more appropriate in this case. Please apply this throughout the manuscript.

5. Line 79-80: ‘All these results will provide information about screening for incRNAs targets in silico.’ Suggestion: If the authors refer to the published results, this sentence should not be in future tense. The authors do not italicize all the terms, such as in silico and in vivo throughout the manuscript.

6. Line 99: It should be ‘development stages’, instead of ‘developing stages’.

7. Line 138-139: ‘The protein-coding genes used were derived from protein coding transcripts in this study and oil palm predicted gene models downloaded from NCBI (Version: EG5).’ Suggestion: The protein-coding transcripts identified in this study were annotated based on the oil palm gene prediction in NCBI (Version: EG5).

8. Line 218-220: This sentence is very confusing. I am assuming that the authors are referring to the ratio of lncRNA to mRNA. Please rephrase it.

9. Line 218, 282, 289: Correlation is wrongly spelled. Please split ‘oilbody’.

In addition, the abbreviations should be defined when they are introduced at the first time. But, the authors do not comply that (e.g. FPKM).

I agree that the findings of lncRNAs and their target genes for mesocarp oil content and fatty acid components can be beneficial to oil palm research. However, the existing literature support is still insufficient to provide a good context/background. For example, the authors mentioned few reports for oil palm (line 69). Please provide the citations. I only found one report for oil palm, coconut and date palm published by Yong Xiao et al. 2019. Many QTL mapping for oil yield-related traits and fatty acid components in oil palm using linkage and GWAS methods were published. The authors can cite these papers to explain how this study can fill up the research gaps. I hope the authors can enhance this with clearer research questions to justify the study. It is also unusual to include the results/findings (line 89-94) in the introduction. Indeed, specifying the study objectives here will be more informative. I observed the same things in discussion. Basically, the authors keep reiterating the results without solid interpretations. For instance, the authors could compare the oil content-associated loci with the reported QTL positions. The benefits of having different libraries of lncRNAs across tissues. What hypothesis can be derived from the expression changes across development stages in mesocarp? In my opinion, the correlation of expression between lncRNA and target genes do not necessarily require the profiling across development stages. Hence, the authors should discuss the purpose and value of doing that (e.g. the best time for expression profiling?). With these information, the authors can suggest next research phase based on the findings. Nevertheless, I cannot see any of these in the discussion. Lastly, try to minimize ‘… and so on/etc.’ in manuscript writing. Readers will not understand how big the scope is.

Experimental design

I also could not see the link of profiling the genes and lncRNAs in different tissues from 4-year-old mature palm (not sure how many palms) and 1-year-old juvenile palm. The profiles might be confounded by the age effects. Again, the authors should be explaining the reasons of the designs and the error mitigation steps when analysing the data.

Validity of the findings

In general, the Bioinformatics analysis is technically sound. However, the methodology is rather brief (see below).

1. Line 97-102: Dura should be referred as fruit form, rather than cultivar. What is the origin of the assayed palms? In this section, the authors do not explain the origins of the discovery population (200 palms) for association study. Besides, the sampling, such as spear leaf (frond number?), kernel and mesocarp need to be further described, because trait variation in different fronds and distribution of fruitlets in a bunch can be significant.

2. Line 183-193: As mentioned, the authors need to explain the source of the discovery populations. I cannot find the description of phenotyping method i.e. quantification of fatty acids and mesocarp oil content.

3. Figure 5: For better illustration, I would suggest changing the heatmaps of differential expression to a heatmap of correlation between lncRNAs and their target genes.

Additional comments

The finding is able to provide a novel way to study the underlying mechanisms for lipid synthesis pathway in oil palm, apart from QTL mapping. However, the authors should strengthen the English writing, introduction with clear experiment questions and discussion to address the questions. The current manuscript is obviously lacking of these elements.

---

## Round 0.2 · Major Revisions

The current version of the manuscript has been significantly improved compared to the previous version. However, not all comments of the reviewers have been addressed. The writing of the manuscript needs improvement to present a more convincing case to the readers. Also, the manuscript needs to be edited by a native English speaker or a technical writer. The reviewers have provided a detailed list of required changes. Please address them very carefully.

Reviewer 1 ·

Basic reporting

This paper improves in many respects on the previous version. Various conclusions not wholly supported by the data have been removed. The methods have also been more completely described, and additional supporting tables added. Nevertheless, I have concerns about the validity of the reporting and do not believe that at present all of my previous comments have been fully addressed.

The paper also still merits further proofreading. There remain numerous spelling and grammatical inconsistences throughout, e.g., "when FPKM values higher than" (line 309), "yeild" (line 433), “collerated” (line 495), and the use of numbers without a thousands separator, e.g. "40 000" (line 98).

(Incidentally, in subsequent sections of this review, line numbers refer to the tracked Word document: the final text with markup shown).

Experimental design

no comment

Validity of the findings

One of my principal concerns from the previous review, and which I do not yet feel is fully addressed, is that many numbers are inconsistently used between the report and the supporting tables.

For instance, line 378 reads "A total of 875 target genes were identified, including natural antisense lncRNAs (NAT-lncRNA, 712), long intergenic noncoding RNAs (lincRNAs, 92), intronic-lncRNAs (33), and sense-lncRNAs (52)".
However, 712 + 92 + 33 + 52 = 889, not 875 (is this discrepancy because lncRNAs are being assigned to more than one category?)
Line 503 re-iterates that there are 712 target genes for NAT-lncRNAs so I assume this is not a one-off typo.
Importantly, this number is not supported by the associated table. Each target gene is detailed in the ‘genomic loci’ column of Table S6, but 694 rows are described with the phrase ‘antisense-overlapped’, not 712 (more specifically, 550 are described as ‘antisense-overlapped’, 100 as ‘antisense-overlapped within 2k downstream region’ and 44 as ‘antisense-overlapped within 2k promoter region’). This figure of 694 is also given on line 383, so where is 712 coming from?

Line 388 - “Comparison of the transcripts of NAT-lncRNAs and their target genes indicated that 262 were overlapped and 288 were not overlapped (Table S6)” – is also incorrect. The table shows that 263 transcripts were ‘antisense-overlapped’, 49 ‘sense-overlapped’, and 616 ‘not overlapped’.

Line 390 – “The lincRNAs were the largest type of identified lncRNAs, we identified 87 target genes for 82 lincRNAs located within 2k regions (Table S6).”
It is not clear what is meant by ‘largest type of identified lncRNAs’. If referring to the number of lncRNAs assigned that category, then Table S6 shows that only 92 of 928 lncRNAs are lincRNAs, so that cannot be right. Has the word ‘largest’ been used instead of ‘longest’? If so, it’s unrelated to the rest of the sentence.
In any case, I find that of the 92 lincRNAs in Table S6, only 69 have an associated gene name, not 87.

Line 336 reads "When FPKM values higher than 15, mRNA (7%) have higher gene ratio than lncRNAs (1%). There were only 23 lncRNA locis had high expression levels (FPKM≥15)."
If 23 genes had FPKM ≥ 15 and this represents 1% of the total number of genes, then it follows that expression has only been quantified for 230 genes - which isn't true. Both sentences are also grammatically incorrect.

Finally, lines 127 to 137. This section states that 18 oil palm transcriptomes were downloaded from NCBI (and refers to Table S1 for complete descriptions of these 18), but then lists 19 accession numbers. DRR053155 (line 129) is missing from Table S1. The manuscript refers to “18 oil palm transcriptomes” throughout; should this be 19?

While individually these are minor issues and some may simply be typos, collectively they greatly confuse interpretation. Many appear to be basic numerical errors that, in conjunction with ambiguous phrasing, make it hard to determine what exactly has been done.

Regarding the lncRNA – target gene pairs on line 204, the Pearson correlation threshold has been lowered from the previous version of the paper (where it was 0.6) and is now 0.468 (on line 376, stated as 0.47). It’s very unusual to have so specific a threshold; I do not believe it is justified (the Zhao et al. Nature Comms paper, used in response to the point about FPKMmax, uses a threshold of 0.6). In any case, this threshold has also not been correctly applied, if it is indeed 0.468: of the 15 lncRNA-target gene pairs in Table S7, one has a Pearson correlation coefficient of 0.466. If numbers are being rounded, they should be presented as such in the table.

One comment from my previous review, and related to this, has not been completely addressed.

Regarding the co-expression analysis to identify lncRNA-target gene pairs, I previously wrote that:

"My concern is that Pearson’s correlation coefficients can be highly skewed by outliers, which is very likely given the low number of datapoints (n=6) and the sparsity of many lncRNA expression profiles (it is expected that of the 6 datapoints per gene, many will be 0; the characteristically low expression of many lncRNAs is already evident from Figure 1B). Notably, the filter criteria described on line 156 (excluding lncRNAs where the maximum FPKM, across all 6 datapoints, is < 1) would not resolve this problem."

To which the reply was:

"LncRNA loci or target genes were excluded for coexpression analysis because of low expression level in 18 transcriptomes (FPKMmax < 1)"

I appreciate that tripling the number of datapoints (to 18) in part addresses some concern here. However, it appears that only lncRNA have been excluded on the basis of FPKMmax < 1 rather than target genes - Table S8 contains target genes for which FPKM is < 1 in all samples. For example, on line 33 of Table S8, MSTRG.15295 supposedly correlates with EgOleosin-1, the highest FPKM of which is 0.67. This “correlation” is strong (r = 0.68) but is not evidently biologically meaningful: the purpose of FPKMmax is to define a threshold below which expression levels are considered indistinguishable from noise. On this basis, oleosin is not detectably expressed in these samples and so the correlation has no validity.
Tangential to this, the paper frequently uses the phrase “low expression level” to refer to FPKMs < 1, such as on line 308. However, on the basis of this threshold, this phrase is misleading - these are not “low” but “negligible”, i.e. not lowly expressed but not detectably expressed at all.
It may also be considered that FPKMmax is a poor choice of threshold as it does not scale with the number of samples. The greater the number of samples, the more likely that any single one will have FPKM > 1 (and so be retained for downstream analysis), irrespective of how many are < 1.
Consider the strongest correlation in Table S7 (line 331), of MSTRG.12655 with LOC105043335. This has r = 0.95 but is made on the basis of a target gene having FPKM of 0 in 16 of the 18 samples, and only one sample above FPKMmax, with an FPKM of 1.28. A positive relationship between the two cannot be a sound conclusion to draw. Accordingly, I doubt the validity of line 478: “Coexpression analysis between lncRNAs and their target genes indicated that 6% lncRNAs have positively correlated expression patterns with target genes”.
A more robust approach would be to consider a % of samples in which FPKM > 1.

The paper does have merit as a survey of lncRNAs in the oil palm and I do appreciate the size and scope of the work. However, I would recommend major revision in order to address the lack of clarity between the text and tables.

Other comments:

Line 100. There is no need to state "with removal rRNA" as this is intrinsic to using RNA-seq.

Line 258. What is meant by "integrity > 0.8"?

Line 336. "close gene proportions for FPKM values between 1 and 15" is a confusing phrase. Also, should "≥ 15" be "> 15", consistent with the "higher than" in the preceding sentence?

Line 478. “median” is used instead of (presumably) “medium”.

It is unusual to include ‘total bases’ per library in Table S1; ‘total number of reads’ would be more informative, as used on line 311.

The highlighted values in Table S7 should be stated as the strongest correlations, according to the threshold given in line 204 (but see comments above).

Table S8 places ** after each correlation but it is not stated what this means (I assume associated p-value < some low number).

"Sheet1" in the Table S8 file contains redundant information. Ditto columns T and U in Table S6.

Supplementary files have been renamed but their contents have not. For instance, the spreadsheets in the Table S2A and Table S2B files are labelled Table S3A and S3B.

Reviewer 3 ·

Basic reporting

I still agree that the findings can be valuable to the oil palm research community, but the current writing can be challenging in conveying the message to the readers. I could see the author’s effort making corrections as pointed out earlier, but errors and format inconsistencies are still easily spotted throughout the manuscript (see below). Again, I would not point out every one of them. Please have another round of careful proofreading and English improvement.

Throughout the manuscript: Inconsistent number separator (comma or spacing) is spotted. The author should refer to the journal format.

Throughout the manuscript: Median is normally referring to the middle point of a range of data. I assume that the author wanted to describe intermediate level, so ‘medium level’ is more proper.

Line 64-66: (Shuai et al. 2014) should be placed after ‘…black cottonwood,’. The citation style should be consistent. The readers will be hard to trace which species is referred by combining multiple citations as one bracket. Besides, Zhang et al. 2014 should worked on rice, instead of wheat.

Line 98: ‘…six developing stages…’ Please change to ‘developmental’.

Line 100: Please remove ‘(8-24W)’ because the six developmental stages are just described in previous sentence. Besides, the readers may misinterpret as one developmental stage between 8 to 24 weeks.

Line 111: Please include a space between ERR1413765 and (100 days of pollination (DAP)).

Line 128: Please change to ‘The fragment size ranged…’.

Line 135 and throughout the manuscript: Please change to ‘FastQC software’. The software should be after the name. Please check the remaining text.

Line 142: Please change to ‘…to the oil palm reference…’.

Line 146: NCBI has been defined at line 90. Please use the acronym here.

Line 159 and Line 162: Please change to ‘E-value cut-off’

Line 180: Change to ‘…used to map the reads on the oil palm reference genome’.

Line 192: Change ‘obove’ to ‘above’.

Line 205: Change to ‘variation in fatty acid composition…’.

Line 22: Change to ‘from an equally…’.

Line 255: Change ‘have’ to ‘had’. Please check throughout the manuscript.

Line 305: Change ‘..’ to ‘.’.

Line 317 and Line 364: Correlation is wrongly spelled again. Please correct it.

Line 317-319: The listing of genes after ‘…with ten lncRNA,’ is improper. Please change to ‘A total of 27 genes in de novo fatty acid synthesis pathway, including EgPDH (5),… were found to be positively correlated with 10 lncRNAs.’

Line 322: Un-space in MSTRG.15295. Please check throughout the manuscript.

Line 324: Please italicise ‘de novo’.

Line 338: Please change to ‘…some lncRNAs were validated and play critical roles…’.

Line 364-365: Please be consistent in naming WRINKED1 (WRI1). Please use acronym WRI1 in line 365 since the author has defined it earlier.

Line 370: Please change to ‘…we found that 77.8% - 89.6%...’.

Line 373: Please use SNP since the acronym has been defined earlier.

As for introduction, some literatures regarding QTL mapping for fatty acid compositions of oil palm were added, but more like a ‘shopping list’. The authors do not really explain further what are the research questions/gaps and how this study can address them. I only observed a sentence of ‘few reports on genome-wide lncRNAs identification in oil palm are available’ at Line 68. Please provide the citation, too. A suggestion here. Many QTL mapping for oil yield, fatty acid composition (including the author’s previous GWAS) and vegetative traits in oil palm are reported. Researcher would require a good reference of oil palm genome, transcriptome profile and lncRNA profile in order to identify the causality for phynotypic variations (fatty acid composition is just one of them). However, the understanding of genome-wide lnRNA is still limited. Same goes to discussion. The author should further compare his findings with the previous reports. For example, how similar QTLs for fatty acid composition across populations reported by Montoya et al. (2013), Singh et al. (2009), Jeennor & Volkaert (2014) and Xia et al. (2019a)? All these should be discussed instead of keep re-iterating that lncRNA is important and played critical roles in many biological functions. Before that, the author needs to address my questions stated in experimental design.

Experimental design

For plant materials, the germplasm garden at Wenchang is not the real origin of the selected palm. The origin of oil palm should be known with native country/research centre, such as Deli dura and AVROS pisifera. I cannot find the information in Xia et al. (2019a), as well. This information is important. Besides, the author has yet explained why choosing root sample from one-year-old seedlings (how many seedlings?), instead of ten-year-old palm. Besides, the pooling of RNA samples is still unclear. Do you mean pooling of triplicate samples per tissues per developmental stage? Or, pooling of different developmental stages of mesocarp? Or, polling of different tissues? This has to be clear.

In addition, the author also needs to explain what is the objective of studying other tissues, such as female inflorescent, kernel and root in the discussion. It seems to be hanging and why only concentrating on mesocarp? Please give your explanation.

Validity of the findings

After knowing the source of discovery population (200 palms) used for association study, I would like to know why the author repeated the association analysis using the SNPs residing on lncRNA regions. The structure/kinship correction for association analysis can be different between the genome-wide and lncRNA levels. Hence, for fair comparison, the author can adopt the same p-value of GWAS reported in Xia et al. (2019a) and follow the FDR-adjusted threshold at –log10 p = 7.3. The number of significant SNPs on lncRNA regions probably can be further reduced and I think that can be a better outcome. Please explain the reason.

Line 331-332: It should be 14 lncRNA loci indicated negative correlation, right?

Additional comments

The current manuscript is not ready for publication. The author is required to proofread the manuscript carefully and to address the questions and comments.

---

## Round 0.3 · Major Revisions

Unfortunately, I have to keep this manuscript in the major revision category. There are numerous technical issues with the described analysis. African Oil Palm genome is not fully assembled therefore the bioinformatics analysis pipelines need to be adjusted to reflect this. Please consider the reviewer's report very carefully and address all concerns.

Reviewer 1 ·

Basic reporting

no comment

Experimental design

no comment

Validity of the findings

no comment

Additional comments

This paper has been improved since previous versions, although I do still have several outstanding concerns.

I have reservations about the use of ‘low or undetectable expression’ (e.g. line 382): ‘undetectable’ is rather distinct from ‘low’. This is important because you are referring to lncRNAs novel to this study, not to genes with pre-existing support. The data used to assemble a given lncRNA and provide evidence for its existence (i.e. RNA-seq reads) is the same as that used to conclude it is undetectable. It cannot be both. If an lncRNA exists such that it may be confidently assembled (that is, is not a methodological artefact), it will also by definition be detectably expressed, on the basis of a given FPKM threshold. If expression is below a given threshold, we will have to discard the lncRNA as a spurious assembly.

This is relevant because there are clearly assembly artefacts in the paper.
Sort the ‘length’ column of Table S4 and you will see some ‘lncRNAs’ of enormous length. Most strikingly, MSTRG.26928 is almost half a million bases long. This is clearly not a contiguous, bona fide, transcript at all but an assembly artefact – it is far more likely that a read or two have been mapped in one location and their mates to a distant one, with the algorithm erroneously reconstructing a transcript model that connects them. To put this figure in context, this transcript is about 35x longer than that derived from one of the largest genes known, human dystrophin (the principal transcript of which is c. 14kb long). Put simply, ‘undetectable expression’ in this case is not a statement that can reasonably be made about MSTRG.26928, as that statement presumes the independent existence of the transcript. I would conclude that given the evidence presented, MSTRG.26928 does not actually exist precisely *because* it is undetectable.

I do appreciate that the content of the supplementary tables is now more helpful (the ‘no. of transcriptomes in which FPKM > 1’ is particularly useful) in that it does easily allow the reader to narrow their focus to the most plausible candidates – and it is not unexpected in a study focusing on lncRNAs that some artefacts will be present. The presence of individual artefacts is not exceptionally concerning *in the tables alone* as they can now be quickly identified.
My two major concerns for publication purposes are where high-level conclusions have been drawn on the basis of a pool of lncRNAs containing spurious members, and where explicit conclusions have been drawn on the basis of named lncRNAs. This necessitates a particularly high standard of proof. In several instances, I would argue that the conclusions about particular lncRNAs do not follow from the data and would revise the discussion accordingly.

For example, line 383 states that MSTRG.17117 positively correlates with 9 genes present on a pathway involved in de novo fatty acid synthesis. These correlations are spurious, so this statement should be removed. MSTRG.17117 is an anomalous 126,203bp long (Table S4) and is now (correctly) flagged as being ‘low expression’ in Table S7, both the ‘lncRNA’ and the ‘target gene’ for this correlation having an FPKMmax < 1. MSTRG.17117 is evidently an artefact and the correlation untenable. While this is now acknowledged in the table, it has yet to be addressed in the text.
The same is also true of MSTRG.16345 (which line 383 states correlates with 5 genes), although this has 0 FPKM in all 18 samples of Table S7 – which contradicts Table S4, in which an averaged FPKM of 0.378 is given. Has there been an issue with rounding in S7 here? Further, in table S7, ‘low expression’ should more accurately be stated as ‘no detectable expression’, ‘negligible expression’, ‘FPKMmax < 1’ or equivalent – for something to be lowly expressed we must first assume there is sufficient evidence that it exists (as discussed above).

Unfortunately the paper remains a challenging read precisely because individual examples like these – which are not difficult to find – cast doubt as to the validity of the body of work as a whole. This is problematic because many of the lncRNAs reported do have a sound evidential basis – but as the discussion occasionally muddles together those which are plausible with those which are so clearly artefactual, I do not yet feel it possible to accept the paper as a complete package.

One particular frustration is that the discussion cites numbers that are not easy to corroborate in the tables without making assumptions about what is intended. For instance, lines 300-304 describe Table S7 as “585 pairs of lncRNA loci and target genes”, for which 338 are filtered out on the basis of ‘low expression’, leaving 247 lncRNA-target gene pairs for which Pearson correlations are calculated. This is fine. However, line 303 then appears to reduce the number of lncRNA-target gene pairs further: “247 pairs of lncRNAs (222) and corresponding target genes (236)”. What is meant here is “247 lncRNA-target gene pairs (representing 222 unique lncRNAs and 236 unique target genes)” but this isn’t explicitly said.

In this respect, the supplementary tables are not, immediately and unambiguously, representing the in-text descriptions, which makes additional (and unnecessary) work for the reader. Line 328 reads: “Twelve lncRNAs had positively correlated expression patterns, with 29 genes belongs to de novo synthesis of fatty acids (20) and TAG synthesis (9) (Table S7).” However – which are the 12 lncRNAs? Which are the 29 target genes? These are not indicated in Table S7, so I am not able to easily follow up on the reference.

I maintain that this paper is a useful survey of novel lncRNAs in the oil palm but I do not feel I can recommend it at this time. While the supplementary tables are much improved, it is clear from the ‘track changes’ version that the text itself has only been minimally edited. Some of the previous rounds of comments are not therefore adequately reflected in the paper itself. In particular, I believe it necessary to only state by name in the main text those lncRNAs which are not assigned to the ‘low expression’ category of Table S7.

Other comments:

Line 144. There are erroneous brackets in the description of Trimmomatic parameters.

Line 328. ‘belongs’ should be ‘belonging’.

Re: using the number of 875 target genes.
It would be clearer to the reader if your explanation – that these are unique/distinct gene IDs – would be incorporated in the abstract. Line 36, for instance, could read “A total of 875 unique target genes were predicted, including those which were intronic (33), sense (52), intergenic (92)…”

Why are there cells marked ‘#N/A’ in Table S7? (e.g. for the expression of MSTRG.7108).

Many of my previous comments regarding methodology have been referred to Zhao, et al. (2018), the informatics aspects of which are largely approximated. However, leaving aside their far larger sample size, there are several reasons why Zhao, et al. produced more robust results and do not have so much of a problem with assembly artefacts:
1. Zhao et al. filter out transcripts on the basis of FPKMmax < 1 BEFORE assessing any coding potential. This avoids the logic problem of assembling a set of putative lncRNAs and then ruling that some are ‘undetectable’ – something must have been detectable otherwise you wouldn’t have been able to assemble them in the first place. In the case of several lncRNAs in the current paper what was detectable was only transcriptomic noise and the assemblies artefactual, as discussed above.
2. Zhao et al. filtered out any transcript with reasonable sequence similarity to any member of a large database of protein-coding genes, SwissProt. The current paper filters out only ‘annotated protein-coding genes’ in oil palm (line 153) and blasts only against a database of transposable elements.
3. Zhao et al. used more stringent read-cleaning criteria; those applied by Trimmomatic in the current paper are rather lenient when it comes to end-trimming (Phred < 3) and minimum read length (36bp).
4. Zhao et al. filtered on the basis of Cufflinks class codes before assessing any coding potential, rather than taking all possible Cufflinks transcripts.

---

## Round 0.4 · Minor Revisions

You have significantly improved the manuscript. In the current revision, the tile and abstract have been appropriately modified,.
The bulk of the analysis was conducted for one subset of all lincRNA (NAT-lncRNA)- this should be made clear and in abstract as well as throughout the manuscript “NAT-lncRNA” should be used where these are being referred to, which is everything relating to targets and expression (i.e. most of the biologically informative reporting) and it is misleading to simply state “lncRNA” in that case. Please check the spelling and address all issues identified by the reviewers before resubmission.

Reviewer 1 ·

Basic reporting

See "general comments" section.

Experimental design

See "general comments" section.

Validity of the findings

See "general comments" section.

Additional comments

Throughout the course of reviewing this paper, my principal criticisms have regarded presentation and methodology: disparities between the text and the tables, and of discussion points raised on the basis of spurious lncRNAs. I now consider both points to have been addressed; the supplementary tables are now more detailed and the dataset as a whole provides a useful overview of the non-coding transcriptome of the oil palm. For these reasons, I would recommend the manuscript for publication. Nevertheless, the PeerJ criteria require that any recommendation be “acceptance as is” although there remain several outstanding (but minor) grammatical points. As such, I must suggest minor revisions for proofreading.

Examples:

Line 34. No need for the word "released"

Lines 36 and 319. "uniq" should be "unique"

Line 91. “gigbases” should be “gigabases”

Line 95. “crop” should be “crops”

Line 98. “to indentify” should be “in identifying”

Line 111. “…which were originated from Malaysia, grown…” should be “…sourced from Malaysia, and grown…”. Rather than starting the sentence with “The African oil palm plants…” please state how many: “X African oil palm plants…”

Line 150. The ENA URL is not necessary here as it is also given in the following line.

Line 163. “ploy-N” should be “poly-N”

Line 165. Please state the version numbers of HISAT2 and StringTie.

Line 184. “fickett” should be “Fickett” and “transcipts” should be “transcripts”.

Line 206. “RNA-seq Sequence Read Archives (SRAs)” is used inappropriately; this is the name of the archive itself, not a file format. “RNA-seq libraries” would do. Change “to calculate” to “and used to calculate”.

Line 216. It is unclear what “six tissues (18) described above” refers to – do you mean “six tissues of the 18 described above”?

Line 251. Please state the version numbers of STRUCTURE and SPAGeDi.

Line 263. “96 009” should be “96,009”. Other numbers used throughout the paper should also be correctly formatted.

Line 402. Arabidopsis should be italicised.

Line 409. “transcriptome” should be “transcriptomes”.

Line 410. “were tended” should be “tended”.

Table S3. The “code” column is confusingly labelled – it is just a number, and does not seem necessary. “Samples” should be “sample name” and a note appended to describe them – they are otherwise opaque. Lines 246-247 imply that these sample names specify individual name and both biological and technical replicate number, but this should be made explicit.

·

Basic reporting

The revised manuscript still has some typo. errors e.g. "uniq"

Experimental design

No comment

Validity of the findings

No comment

Additional comments

With the modifications to title and inclusion of the FA analysis data, the manuscript is improved. Please again carefully check spelling (e.g. “uniq”). For precision and clarity, please use “NAT-lncRNA” not just “lncRNA” where the data is only based on the NAT-lncRNA (which is actually in most cases of use).

---

## Round 0.5 · accepted · Accept

Thank you very much for implementing all the suggested changes. Your work is important for scientists working in the oil palm biotechnology area.